# Host casein kinase 1-mediated phosphorylation modulates phase separation of a rhabdovirus phosphoprotein and virus infection

Xiao-Dong Fang[1], Qiang Gao[1,2], Ying Zang[1], Ji-Hui Qiao[1], Dong-Min Gao[1], Wen-Ya Xu[1], Ying Wang[2], Dawei Li[1], Xian-Bing Wang[1]*

[1]State Key Laboratory of Agro-Biotechnology, College of Biological Sciences, China Agricultural University, Beijing, China; [2]College of Plant Protection, China Agricultural University, Beijing, China

*For correspondence:
wangxianbing@cau.edu.cn

Competing interest: The authors declare that no competing interests exist.

**Abstract** Liquid-liquid phase separation (LLPS) plays important roles in forming cellular membraneless organelles. However, how host factors regulate LLPS of viral proteins during negative-sense RNA (NSR) virus infection is largely unknown. Here, we used *barley yellow striate mosaic virus* (BYSMV) as a model to demonstrate regulation of host casein kinase 1 (CK1) in phase separation and infection of NSR viruses. We first found that the BYSMV phosphoprotein (P) formed spherical granules with liquid properties and recruited viral nucleotide (N) and polymerase (L) proteins in vivo. Moreover, the P-formed granules were tethered to the ER/actin network for trafficking and fusion. BYSMV P alone formed droplets and incorporated the N protein and the 5′ trailer of genomic RNA in vitro. Interestingly, phase separation of BYSMV P was inhibited by host CK1-dependent phosphorylation of an intrinsically disordered P protein region. Genetic assays demonstrated that the unphosphorylated mutant of BYSMV P exhibited condensed phase, which promoted viroplasm formation and virus replication. Whereas, the phosphorylation-mimic mutant existed in diffuse phase state for virus transcription. Collectively, our results demonstrate that host CK1 modulates phase separation of the viral P protein and virus infection.

## Editor's evaluation

This paper reveals that the phosphoprotein (P) of a plant negative-sense RNA virus, Barley yellow striate mosaic virus (BYSMV), forms condensates through liquid-liquid phase separations (LLPSs). Using BYSMV minireplicon system, they show that the unphosphorylated P protein undergoes phase separation to promote virus replication. Whereas, the host casein kinase 1 (CK1) phosphorylates P protein to inhibit phase separation and virus replication.

## Introduction

Over the last few years, increasing studies have shown that liquid-liquid phase separation (LLPS) has critical roles in assembly of cellular membraneless organelles such as P bodies, stress granules, cajal bodies, and the nucleolus (*Handwerger et al., 2005*; *Brangwynne et al., 2009*; *Molliex et al., 2015*; *Feric et al., 2016*; *Boeynaems et al., 2018*; *Darling et al., 2019*). LLPS concentrates specific molecules like proteins and nucleic acids into liquid-like compartments for fulfillment of their biological functions. The underlying molecular mechanisms have been of increased interests due to the important roles of LLPS in various physiological and pathological processes (*Darling et al., 2019*). LLPS is usually

triggered by intrinsically disordered regions (IDRs) of proteins and/or multivalent macromolecular interactions (*Elbaum-Garfinkle et al., 2015*; *Molliex et al., 2015*; *Alberti et al., 2018*). In addition, LLPS is modulated by protein posttranslational modifications, host factors, and cellular environment changes (*Nott et al., 2015*; *Banani et al., 2017*; *Owen and Shewmaker, 2019*).

Many negative-sense RNA (NSR) viruses are known to form membraneless replication compartments, called viroplasms, viral inclusion bodies (IBs), or viral factories (*Lahaye et al., 2009*; *Hoenen et al., 2012*; *Rincheval et al., 2017*). Studies about animal NSR viruses have revealed that LLPS plays important roles in viroplasm formation through concentrating viral and host components. The viroplasms of the rabies virus known as Negri bodies (NBs) were first reported to have the features of liquid organelles (*Nikolic et al., 2017*). Subsequent studies have shown that another two animal NSR viruses, vesicular stomatitis virus and measles virus, also exploit LLPS to form virus IBs (*Heinrich et al., 2018*; *Zhou et al., 2019*). The P protein of borna disease virus and the N protein of ebola virus are also sufficient to elicit formation of liquid organelles alone (*Charlier et al., 2013*; *Miyake et al., 2020*). However, most of these studies mainly focus on animal viruses (*Brocca et al., 2020*; *Su et al., 2021*), whereas it remains very limited in plant viruses. Recently, Li et al. revealed that turnip mosaic virus, a positive-stranded RNA virus, hijacks host RNA helicase proteins to form viral bodies through LLPS with viral proteins for viral proliferation (*Li et al., 2021*). In addition, the long-distance movement protein p26 of pea enation mosaic virus 2 undergoes phase separation with cellular factors to modulate virus-host interactions (*Brown et al., 2021*). Nonetheless, host factors regulating LLPS of plant NSR viroplasms are still largely unknown.

*Barley yellow striate mosaic virus* (BYSMV) is a member of the *Cytorhabdovirus* genus, family Rhabdoviridae in the order Mononegavirales. BYSMV infects cereal plants and severely affects crop production worldwide through persistent transmission by the small brown planthopper (*Laodelphax striatellus*). The BYSMV genome encodes five structural proteins, including the nucleoprotein (N), phosphoprotein (P), matrix protein (M), glycoprotein (G), and polymerase (L), as well as another five accessory proteins, in the order 3′–N–P–P3–P4/P5–P6–M–G–P9–L–5′ (*Yan et al., 2015*). Recently, we have developed minireplicon (BYSMV-antigenomic MR [agMR]) systems and full-length cDNA clones of recombinant BYSMV (rBYSMV) for infections of plants and insects (*Fang et al., 2019*; *Gao et al., 2019*). Using the BYSMV reverse genetic systems, we have also shown that host factors, including casein kinase 1 (CK1) and the deadenylation factor CCR4, are involved in virus cross-kingdom infections of host plants and insect vectors (*Gao et al., 2020*; *Zhang et al., 2020*). Interestingly, CK1-mediated phosphorylation of a highly serine-rich (SR) motif at the C-terminal IDR of the P protein regulates virus infection (*Gao et al., 2020*). However, the mechanisms underlying regulation of the BYSMV P phosphorylation in virus infection are not well understood.

Here, using live-cell fluorescence microscopy to observe the localization of the BYSMV N, P, and L core replication proteins, we noticed that ectopic expression of the BYSMV P protein alone resulted in spherical cytoplasmic granules. We also found that the BYSMV P-formed granules have properties of liquid organelles in vivo and in vitro. Host CK1-mediated phosphorylation of BYSMV P inhibits LLPS. The roles of the BYSMV P phosphorylation and the LLPS in virus replication and transcription were investigated and discussed.

## Results

### The BYSMV P protein forms liquid-like granules through LLPS in vivo

To observe BYSMV viroplasms in vivo, BYSMV-infected barley stems were cut into ultra-thin sections and the structures formed in the cytoplasm were monitored by transmission electron microscopy (TEM). In agreement with animal NSR viruses (*Hoenen et al., 2012*; *Rincheval et al., 2017*), BYSMV infection induced formation of cytoplasm inclusions containing condensed ribonucleoproteins (RNPs) (*Figure 1A*). We further performed immunoelectron microscopy using BYSMV P antibodies and demonstrated that gold particles specifically labeled the cytoplasm viroplasm (*Figure 1—figure supplement 1*).

During rhabdovirus infection, the viroplasms mainly consist of the N, P, and L proteins for replication and/or transcription (*Jackson et al., 2005*). To determine the core proteins involved in eliciting viroplasm formation, we examined the subcellular localization of ECFP-N, GFP-P, and L-mCherry in *N. benthamiana* leaves. At 2 days post infiltration (dpi), confocal imaging revealed that only GFP-P,

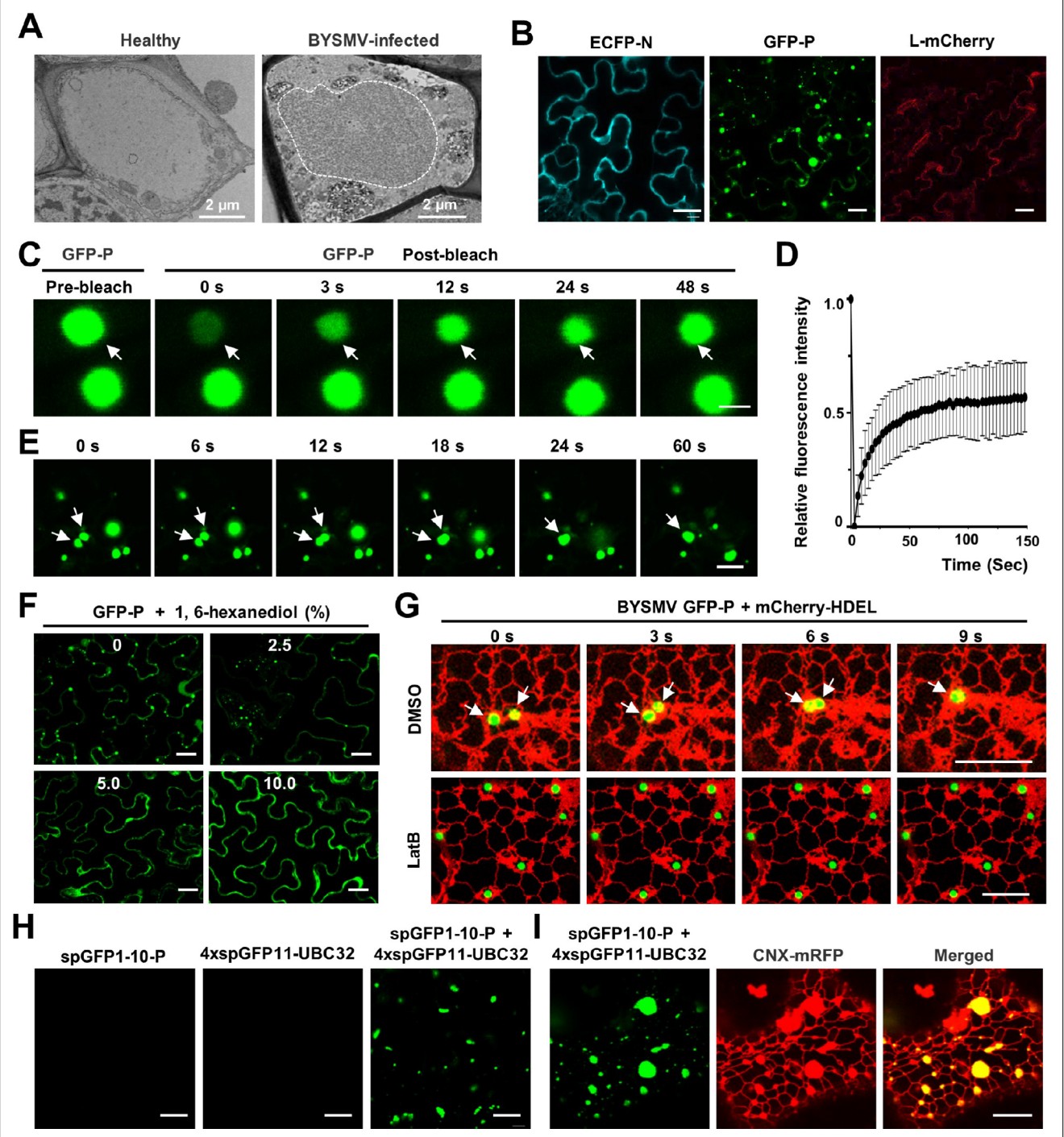

**Figure 1.** Barley yellow striate mosaic virus (BYSMV) phosphoprotein (P) protein forms liquid-like granules through liquid-liquid phase separation (LLPS) in vivo. (**A**) Transmission electron microscopy characterization of the BYSMV viroplasm in barley stems infected by BYSMV at 10 days post infiltration (dpi). The electron dense granular structure of the BYSMV viroplasm is highlighted by white dotted line. Healthy stems served as negative control. Scale bars, 2 μm. (**B**) Confocal images showing subcellular distribution of ECFP-N, GFP-P, and L-mCherry in epidermal cells of *Nicotiana benthamiana* leaves at 2 dpi. Scale bars, 20 μm. (**C**) Representative fluorescence recovery after photobleaching (FRAP) images of GFP-P granules in epidermal cells of *N. benthamiana* leaves at 2 dpi. Leaves were treated with 10 μM latrunculin B (LatB) to inhibit movement of GFP-P granules at 3 hr before photobleaching. Scale bar, 2 μm. (**D**) FRAP recovery curves of GFP-P granules. The intensity of each granule was normalized against their pre-bleach fluorescence. Data were presented as mean ± SD of 15 granules. (**E**) Confocal images showing fusion of two GFP-P granules in *N. benthamiana* leaf epidermal cells. White arrows indicate that GFP-P granules undergo fusion. Scale bar, 10 μm. (**F**) Representative images showing GFP-P localization after treatment with 0, 2.5%, 5.0%, 10.0% of 1,6-hexanediol for 5 min in *N. benthamiana* leaf epidermal cells. Scale bars, 20 μm. (**G**) Time-lapse confocal micrographs showing

*Figure 1 continued on next page*

*Figure 1 continued*

the localization of GFP-P and mCherry-HDEL expressed in *N. benthamiana* leaf epidermal cells at 2 dpi. Leaves were treated with DMSO or 10 μM LatB at 3 hr before imaging. White arrows indicate fusion of two GFP-P granules. Scale bars, 10 μm. (**H**) Confocal micrographs of *N. benthamiana* leaf epidermal cells expressed spGFP1-10-P, 4× spGFP11-UBC32, or both at 2 dpi. Scale bars, 20 μm. (**I**) Confocal micrographs of *N. benthamiana* leaf epidermal cells spGFP1-10-P, 4× spGFP11-UBC32, and CNX-RFP at 2 dpi. The green signals indicate the contact sites of spGFP1-10-P granules with the tubular ER network. CNX-RFP is an ER marker. Scale bar, 10 μm.

The online version of this article includes the following video, source data, and figure supplement(s) for figure 1:

**Source data 1.** Fluorescence intensities of the bleached droplets during the time course experiment.

**Figure supplement 1.** Immunoelectron microscopy detecting the barley yellow striate mosaic virus (BYSMV) phosphoprotein (P) protein in BYSMV-RFP-infected stems.

**Figure supplement 2.** Barley yellow striate mosaic virus (BYSMV) phosphoprotein (P) forms spherule granule recruiting nucleotide (N) and polymerase (L) proteins.

**Figure supplement 3.** GFP-P moved along the actin filaments.

**Figure supplement 4.** GFP-P P forms liquid structures on ER network.

**Figure supplement 5.** Phase separation of phosphoprotein (P) protein in maize and barley protoplasts.

**Figure supplement 6.** Northern cereal mosaic virus (NCMV) phosphoprotein (P) forms liquid structures in vivo.

**Figure supplement 6—source data 1.** Fluorescence intensities of the bleached droplets during the time course experiment.

**Figure 1—video 1.** Trafficking and fusion video of barley yellow striate mosaic virus (BYSMV) GFP-P graunles in epidermal cells of *Nicotiana benthamiana* leaves at 2 days post infiltration (dpi).
https://elifesciences.org/articles/74884/figures#fig1video1

**Figure 1—video 2.** Representative fluorescence recovery after photobleaching (FRAP) video of barley yellow striate mosaic virus (BYSMV) GFP-P granules transiently expressed in epidermal cells of *Nicotiana benthamiana* leaves at 2 days post infiltration (dpi).
https://elifesciences.org/articles/74884/figures#fig1video2

rather than ECFP-N or L-mCherry, formed spherical granules throughout the cytoplasm (*Figure 1B*; *Figure 1—video 1*). Furthermore, ECFP-N and L-mCherry proteins were recruited into GFP-P spherical granules, whereas CFP-N and L-mCherry failed to form condensates with free GFP (*Figure 1—figure supplement 2*). These results suggest that BYSMV P is the scaffold factor for formation of spherical granules and then recruits the N and L proteins into condensed RNPs to facilitate viral infection.

We next used fluorescence recovery after photobleaching (FRAP) to determine whether the BYSMV GFP-P spherical granules have liquid properties. After photobleaching, approximately 56.7% of GFP-P granule signal gradually recovered within 150 s (*Figure 1C and D*; *Figure 1—video 2*), indicating a rapid redistribution of the GFP-P protein between the membraneless granules and the surrounding cellular proteins. In addition, these GFP-P granules moved in the cytoplasm to fuse with each other (*Figure 1E*; *Figure 1—video 1*), and treatment with 1,6-hexanediol (HEX), a chemical inhibitor of liquid-like droplets, efficiently dispersed the GFP-P granules (*Figure 1F*). Thus, these results demonstrate that the BYSMV P protein forms liquid-like granules through LLPS in vivo.

Emerging evidence shows that membrane-bound organelles provide platforms for assembly, fusion, and transport of membraneless granular condensates (*Lee et al., 2020*; *Zhao and Zhang, 2020*). To further evaluate the spatiotemporal dynamics of the GFP-P granules, mCherry-HDEL, a fluorescent ER marker, was monitored by fluorescence microscopy. Time-lapse confocal imaging analyses showed that GFP-P granules are tethered tightly to the ER network and that their dynamics were correlated over time (*Figure 1G*). Treatment with the actin-depolymerizing agent latrunculin B (LatB) reduced both ER streaming and trafficking of BYSMV-P granules (*Figure 1G*). Time-lapse confocal analyses also consistently showed that GFP-P granules moved along actin filaments marked by GFP-ABD2-GFP (*Figure 1—figure supplement 3*). These results suggest that the BYSMV-P granules move rapidly and fuse with each other during ER streaming in an actin-dependent manner.

Trafficking of GFP-P granules in close association with ER tubules strongly suggested that the GFP-P granules were tethered to ER tubules at molecular distances (10–30 nm) as membrane contact sites (*Phillips and Voeltz, 2016*). To examine the extent to which BYSMV P granules are tethered to the ER tubules, we used dimerization-dependent fluorescent protein domains to resolve the nanoscale resolution of GFP-P-ER contact in living cells. Previous studies have shown that two split GFP super-folder components (spGFP1-10 and spGFP11) form functional spGFP green fluorescent signals when the two components interact at molecular distances (*Pédelacq et al., 2006*; *Pedelacq and Cabantous,*

*2019*). We fused a truncated form of the ER-localized ubiquitin conjugase UBC32 with 4× spGFP11 as an ER contact site marker (4× spGFP11-UBC32) (*Cui et al., 2012*; *Li et al., 2020*), and also fused the spGFP1-10 to the N terminus of P (spGFP1-10-P). At 2 dpi, neither 4× spGFP11-UBC32 nor spGFP1-10-P produced GFP signal when expressed alone (*Figure 1H*). Only co-expression of spGFP1-10-P and 4× spGFP11-UBC32 formed GFP-labeled bodies that overlapped with the ER marker, CNX-RFP (*Figure 1H and I*; *Li et al., 2020*). By contrast, co-expression of spGFP1-10-P and LRR84A-GS-2× spGFP11 (*Li et al., 2020*), a plasma membrane marker, reconstituted GFP fluorescence but not in bodies (*Figure 1—figure supplement 4*). Collectively, these results indicate that the GFP-P granules were formed through tethering to the tubular ER network at molecular distances.

Collectively, our results suggest that the BYSMV P protein forms liquid spherical granules through LLPS in vivo. Furthermore, the ER/actin network provides a platform for dynamics of BYSMV-P granules. In consistence, GFP-P, rather than GFP, underwent phase separation and formed granules in protoplasts of maize and barley, as well as protoplasts isolated from rBYSMV-RFP-infected barley leaves (*Figure 1—figure supplement 5*). In addition, the GFP-P protein of a closely related *Cytorhabdovirus*, *Northern cereal mosaic virus*, had similar of liquid spherical granule features (*Figure 1—figure supplement 6*).

## BYSMV P undergoes phase separation in vitro

To determine whether GFP-P liquid spherical granules are directly affected by the BYSMV P protein, we performed in vitro experiments to test phase separation of purified BYSMV P. To this end, we purified the recombinant proteins GFP and GFP-P from *Escherichia coli* (*Figure 2—figure supplement 1A*). As expected, the GFP-P protein, but not GFP, was able to condense into spherical droplets (*Figure 2A*; *Figure 2—figure supplement 1B*). Moreover, treatment with HEX (5%) efficiently dispersed the GFP-P droplets (*Figure 2B*). Moreover, increased GFP-P concentration and decreased NaCl concentration enhanced numbers and sizes of GFP-P droplets (*Figure 2C*).

We further exploited FRAP to quantify molecular dynamics within the GFP-P droplets by showing that ~45% of the GFP-P signal in the droplets gradually recovered within 400 s after photobleaching (*Figure 2D and E*; *Figure 2—video 1*). Note that the FRAP recovery ratio of GFP-P droplets in vitro is less than that of GFP-P granules in vivo (*Figure 1C and D*), indicating that the His tag might affect the FRAP of GFP-P droplets, or other cellular components facilitate phase separation of GFP-P in vivo. In addition, we observed that two approaching GFP-P droplets fused into a bigger droplet (*Figure 2F*; *Figure 2—video 2*). In summary, these in vitro results confirm that the BYSMV P protein alone can undergo phase separation in vitro.

## P-formed droplets recruit the N protein and 5′ trailer of BYSMV genome in vitro

Given that GFP-P granules could recruit the ECFP-N and L-mCherry proteins in vivo (*Figure 1—figure supplement 2*), we next examined whether the BYSMV-P-formed droplets concentrated BYSMV N and genomic RNAs in vitro. The purified mCherry-N alone or with free GFP did not undergo LLPS in vitro (*Figure 3A*; *Figure 2—figure supplement 1B*). However, when mCherry-N was incubated with GFP-P, the mCherry-N protein was gradually incorporated into GFP-P-formed droplets (*Figure 3A*).

To test whether the GFP-P and mCherry-N droplets can recruit genomic RNA, a 334 nt RNA fragment corresponding to the 5′ trailer of BYSMV negative RNA genome was labeled by Cy5 (Cy5-Trailer). The Cy5-Trailer fragment was added to GFP-P/mCherry-N or GFP/mCherry-N mixtures in vitro. As expected, both mCherry-N and Cy5-Trailer were incorporated into the GFP-P droplets, whereas they appeared to be evenly distributed when incubated with GFP (*Figure 3B*). Taken together, these results suggest that BYSMV P-formed liquid droplets can incorporate BYSMV N and the 5′ trailer of BYSMV genome in vitro.

## Phase separation of BYSMV P is inhibited by phosphorylation of the P protein

BYSMV P is a phosphoprotein whose phosphorylation states affect virus replication and transcription (*Gao et al., 2020*). In silico predictions via PONDR suggest that the BYSMV P protein contains three IDRs (*Figure 4A*). Interestingly, five highly phosphorylated Ser residues (amino acids 189, 191, 194, 195, and 198) are present in an SR motif ([189]SASRPSSIAS[198]) located in the middle IDR of BYSMV P (*Gao*

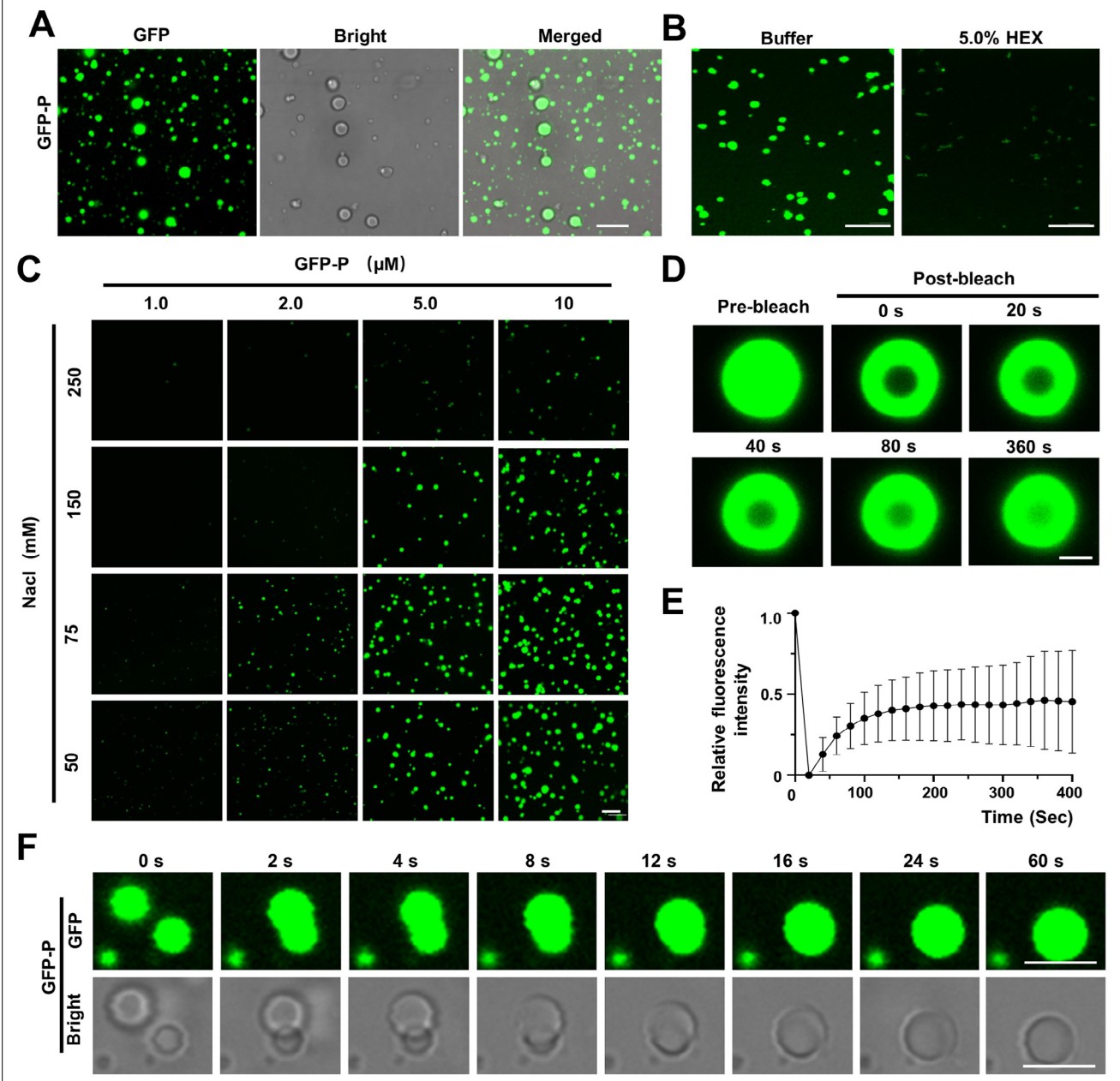

**Figure 2.** Barley yellow striate mosaic virus (BYSMV) phosphoprotein (P) undergoes phase separation in vitro. (**A**) Confocal images showing that GFP-P formed droplets at the concentration of 10 μM in 125 mM NaCl. Scale bars, 10 μm. (**B**) Representative confocal images showing GFP-P droplets before or after treatment with 5.0% of 1,6-hexanediol for 1 min. Scale bars, 10 μm. (**C**) Phase separation of GFP-P at different concentrations of GFP-P and NaCl. Scale bar, 20 μm. (**D**) Representative fluorescence recovery after photobleaching (FRAP) of GFP-P droplets in vitro at the concentrations of 10 μM in 125 mM NaCl. Scale bar, 1 μm. (**E**) FRAP recovery curve of GFP-P droplets. Data are shown as the mean ± SD of 12 droplets. (**F**) Representative images showing fusion of two GFP-P droplets in vitro at the concentration of 15 μM in 150 mM NaCl. Scale bars, 5 μm.

The online version of this article includes the following video, source data, and figure supplement(s) for figure 2:

**Source data 1.** Fluorescence intensities of the bleached droplets during the time course experiment.

**Figure supplement 1.** GFP and barley yellow striate mosaic virus (BYSMV) mCherry-N are deficient in forming liquid droplets in vitro.

**Figure supplement 1—source data 1.** SDS-PAGE showing purified GFP, GFP-P, and mCherry-N proteins (Related to *Figure 2—figure supplement 1A*).

**Figure 2—video 1.** Representative fluorescence recovery after photobleaching (FRAP) video of barley yellow striate mosaic virus (BYSMV) GFP-P droplets in vitro.

https://elifesciences.org/articles/74884/figures#fig2video1

*Figure 2 continued on next page*

*Figure 2 continued*

**Figure 2—video 2.** Representative video showing fusion of barley yellow striate mosaic virus (BYSMV) GFP-P droplets in vitro.
https://elifesciences.org/articles/74884/figures#fig2video2

*et al., 2020*). Given that protein IDRs are usually involved in phase separation (*Brangwynne et al., 2015*), we hypothesized that the phosphorylation states of the BYSMV SR region might affect the phase separation ability of BYSMV P. To test this hypothesis, we carried out site-directed mutagenesis to replace each Ser residue in the middle IDR (IDR2) with an Ala (GFP-P$^{S5A}$) or Asp residues (GFP-P$^{S5D}$) to mimic unphosphorylated and hyperphosphorylated states of GFP-P$^{WT}$, respectively (*Figure 4A*). The GFP-P$^{WT}$, GFP-P$^{S5A}$, and GFP-P$^{S5D}$ proteins were individually expressed in *N. benthamiana* leaves by agroinfiltration. Intriguingly, in contrast to GFP-P$^{WT}$ (~111 granules per view, >0.2 µm²), GFP-P$^{S5A}$ formed relative fewer (~67 granules per view) but larger granules, whereas GFP-P$^{S5D}$ formed very fewer granules (~17 granules per view) and more evenly located in the cytoplasm (*Figure 4B and C*, and large views in *Figure 4—figure supplement 1*). Statistical analyses revealed that the sizes of about 16.4% of the GFP-P$^{WT}$ granules were larger than 2 µm² in diameter, whereas approximately 33.3% GFP-P$^{S5A}$ granules had larger areas (>2 µm²) (*Figure 4D*). In consistence with GFP-P$^{WT}$ (*Figure 1C and D*), the FRAP assays showed that approximately 60% of the GFP-P$^{S5A}$ granule signals gradually recovered within 150 s after photobleaching (*Figure 4—figure supplement 2*).

In previous studies, cellular P bodies can be isolated from animal cells by centrifugation and fluorescence activated particle sorting (*Hubstenberger et al., 2017*). To further examine condensed or diffuse states of GFP-P$^{WT}$, GFP-P$^{S5A}$, and GFP-P$^{S5D}$, total extracted protein samples of infiltrated leaves were centrifuged at 10,000× *g* for 10 min, and the supernatant and pellet fractions were subjected to Western blotting analyses with anti-P antibodies (*Figure 4E*). Interestingly, two bands of GFP-P$^{WT}$ corresponding to about 72 kD (P72) and 74 kD (P74) were present in the samples, whereas GFP-P$^{S5A}$ and GFP-P$^{S5D}$ existed as P72 and P74, respectively (*Figure 4F*). As expected, the GFP-P$^{S5D}$ protein was mainly concentrated in the supernatant fraction, while the GFP-P$^{S5A}$ protein was mainly in the pellet fraction (*Figure 4F*), indicating that GFP-P$^{S5D}$ and GFP-P$^{S5A}$ primarily existed as soluble and condensed states, respectively.

We further examined in vitro phase separation of the purified recombinant GFP-P$^{WT}$, GFP-P$^{S5A}$, and GFP-P$^{S5D}$ proteins (*Figure 4G*). The results were consistent with the in vivo results (*Figure 4B*), as GFP-P$^{WT}$ and GFP-P$^{S5A}$, but not GFP-P$^{S5D}$, underwent phase separation in 150 mM NaCl (*Figure 4H*). Assays at different protein concentrations in 125 mM NaCl indicated that the dephosphorylation state

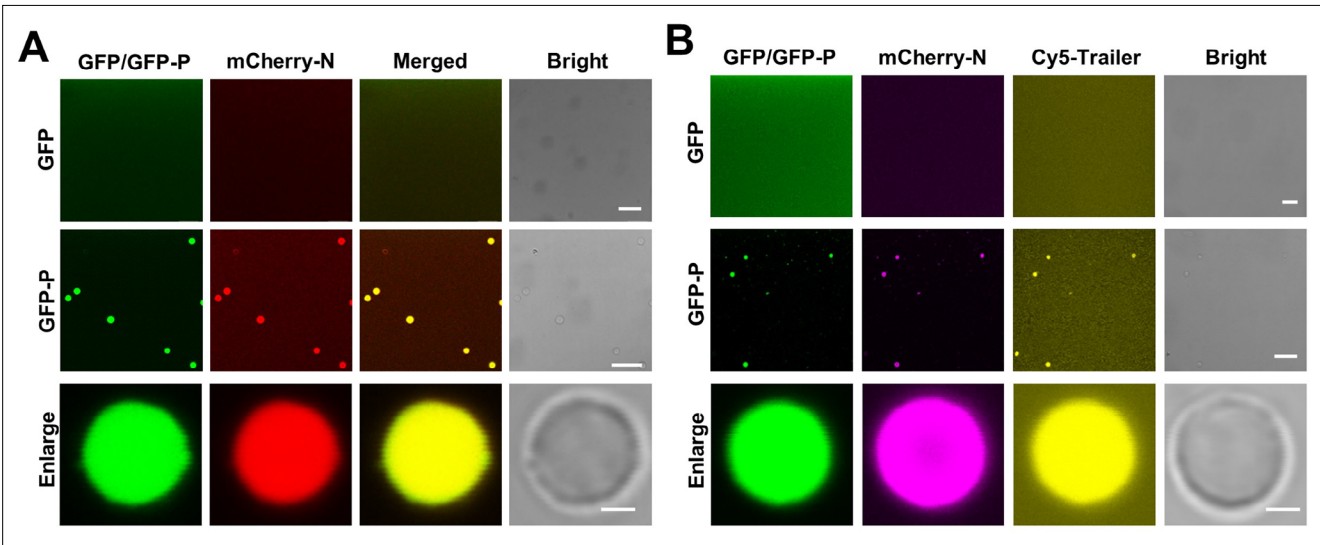

**Figure 3.** P-formed droplets recruit the N protein and genomic RNA in vitro. (**A**) Confocal images showing incorporation of mCherry-N into GFP-P droplets. Free GFP was unable to form droplets to recruit mCherry-N. Scale bars, 20 µm. Scale bars (enlarge panel), 1 µm. (**B**) Confocal images showing incorporation of mCherry-N and Cy5-Trailer of barley yellow striate mosaic virus (BYSMV) genome into GFP-P droplets. In contrast, free GFP was unable to form droplets or recruit mCherry-N and Cy5-labeled trailer. Scale bars, 20 µm. Scale bar (enlarge panel), 1 µm.

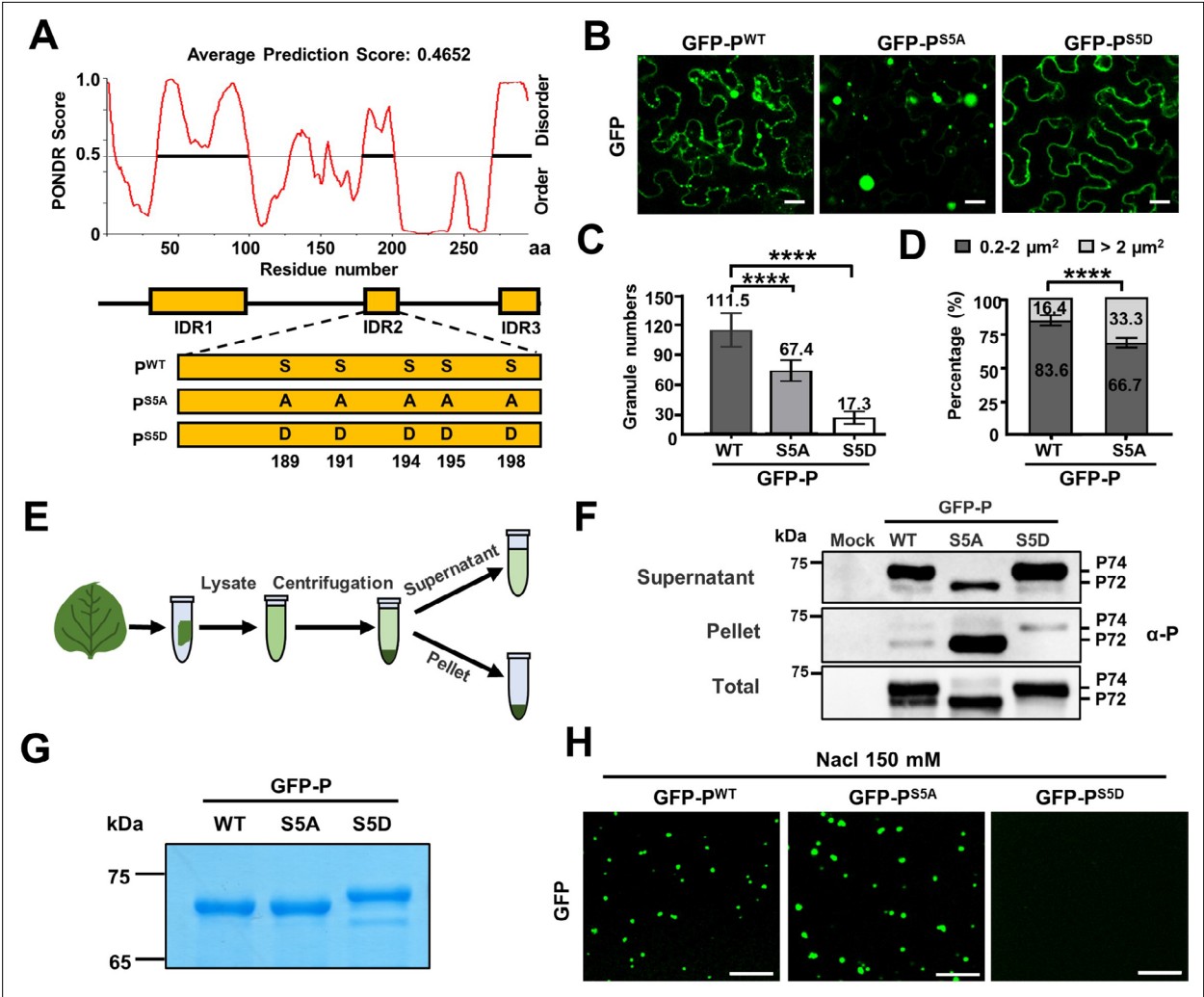

**Figure 4.** Phase separation of barley yellow striate mosaic virus (BYSMV) phosphoprotein (P) is inhibited by P protein phosphorylation. (**A**) The predicted intrinsically disordered regions (IDRs) of BYSMV P and schematic representation of its phosphorylation mutants (P^S5A and P^S5D). IDRs were predicted according to the online tool PONDR and indicated by yellow boxes. (**B**) Confocal images showing subcellular distribution of GFP-P^WT, GFP-P^S5A, and P-GFP^S5D in *Nicotiana benthamiana* leaf epidermal cells at 2 days post infiltration (dpi). Scale bars, 20 μm. (**C**) Statistical analyses of GFP granule numbers (>0.2 μm²) in a field (175 μm × 175 μm) of *N. benthamiana* leaves expressing GFP-P^WT, GFP-P^S5A, or GFP-P^S5D. Error bars indicate SD of eight representative fields. ****p < 0.0001 (Student's t-test). (**D**) Statistical diameter analyses of GFP-P^WT and GFP-P^S5A granules with different sizes (n > 500). (**E**) Workflow showing granule sedimentation assays using *N. benthamiana* leaves expressing GFP-P^WT, GFP-P^S5A, or P-GFP^S5D at 2 dpi. (**F**) Western blotting analyses of supernatant, pellet, and total proteins isolated in panel E. (**G**) SDS-PAGE showing purified GFP-P^WT, GFP-P^S5A, or P-GFP^S5D purified from *Escherichia coli*. (**H**) Confocal images showing droplet formed by GFP-P^WT, GFP-P^S5A, or P-GFP^S5D in vitro. Scale bar, 10 μm.

The online version of this article includes the following source data and figure supplement(s) for figure 4:

**Source data 1.** Statistical analyses of GFP-P^WT, GFP-P^S5A, or GFP-P^S5D granule numbers and diameter (Related to *Figure 4C, D*).

**Figure supplement 1.** Confocal images showing subcellular distribution of GFP-P^WT, GFP-P^S5A, and P-GFP^S5D in *Nicotiana benthamiana* leaves at 2 days post infiltration (dpi).

**Figure supplement 2.** Fluorescence recovery after photobleaching (FRAP) experiment analysis liquid quality of GFP-P^S5A granules.

**Figure supplement 2—source data 1.** Fluorescence intensity of the bleached droplets during the time course experiment.

**Figure supplement 3.** Phase separation of GFP-P^WT and GFP-P^S5A.

**Figure supplement 3—source data 1.** Turbidity assays (OD600) of GFP, GFP-P^WT, and P-GFP^S5A (Related to *Figure 4—figure supplement 3B*).

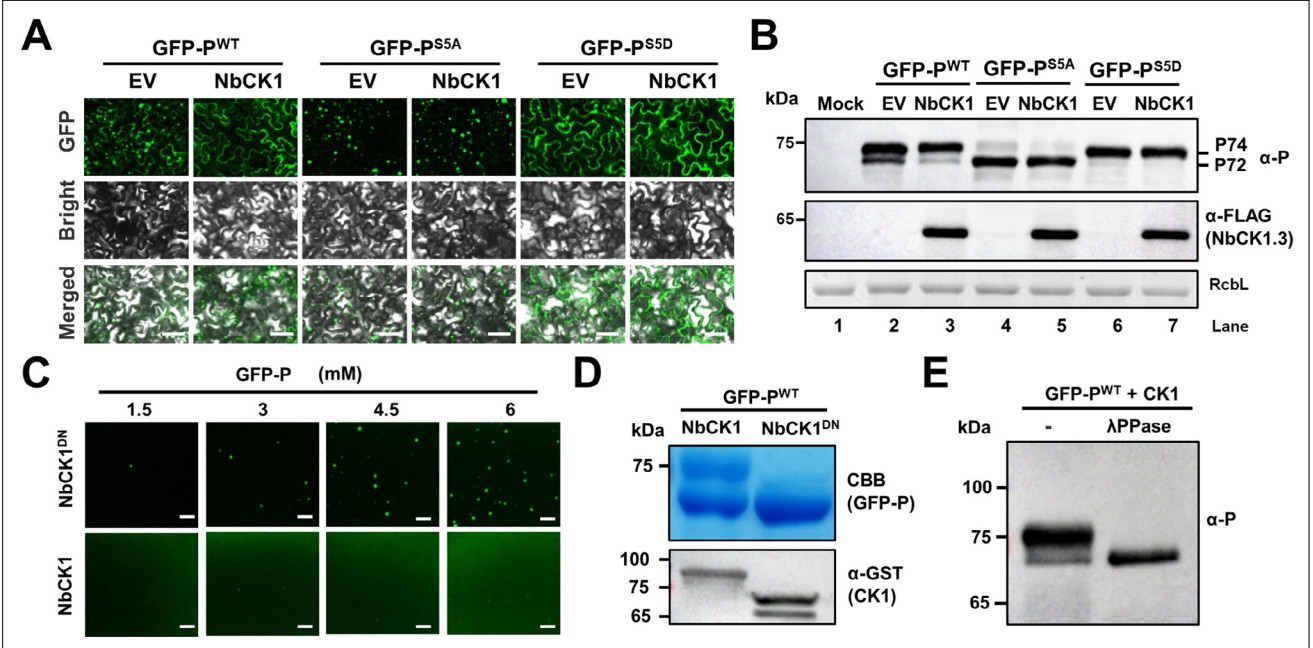

**Figure 5.** Host casein kinase 1 (CK1) inhibits phase separation of barley yellow striate mosaic virus (BYSMV) phosphoprotein (P) in vivo and in vitro. (**A**) Confocal images showing subcellular distribution of GFP-P$^{WT}$, GFP-P$^{S5A}$, and P-GFP$^{S5D}$ co-expressed with empty vector (EV) or NbCK1.3 in *Nicotiana benthamiana* leaf epidermal cells at 2 days post infiltration (dpi). Scale bars, 50 µm. (**B**) Western blotting detecting accumulation of GFP-P$^{WT}$, GFP-P$^{S5A}$, and P-GFP$^{S5D}$ in the leaves as shown in panel A. (**C**) Confocal images showing droplet formation of GFP-P$^{WT}$ purified from *Escherichia coli* co-expressing NbCK1.3 or NbCK1.3$^{DN}$. GFP-P$^{WT}$ was diluted to different concentration and 125 mM NaCl. Scale bars, 20 µm. (**D**) SDS-PAGE showing purified GFP-P$^{WT}$, GFP-P$^{S5A}$, or GFP-P$^{S5D}$ in the samples of panel C. Expression of GST-tagged NbCK1.3 or NbCK1.3$^{DN}$ was examined by Western blotting analyses with anti-GST antibodies. (**E**) Western blot detecting GFP-P$^{WT}$ treated with lambda protein phosphatase ( $\lambda$ -PPase) or mock buffer (-) with anti-P antibodies.

The online version of this article includes the following source data and figure supplement(s) for figure 5:

**Source data 1.** Soure images (Rlated to *Figure 5B, D, E*).

**Figure supplement 1.** Host casein kinase 1 (CK1) inhibits phase separation of barley yellow striate mosaic virus (BYSMV) phosphoprotein (P) in vivo.

**Figure supplement 1—source data 1.** Statistical analyses of GFP-P$^{WT}$, GFP-P$^{S5A}$ and GFP-P$^{S5D}$ granules co-expressed with empty vector (EV) or NbCK1 (Related to *Figure 5—figure supplement 1B*).

**Figure supplement 2.** Host casein kinase 1 (CK1) inhibits phase separation of barley yellow striate mosaic virus (BYSMV) phosphoprotein (P) in vivo.

**Figure supplement 2—source data 1.** Soure images (Rlated to *Figure 5—figure supplement 2B*).

**Figure supplement 3.** Phase separation of barley yellow striate mosaic virus (BYSMV) phosphoprotein (P) is inhibited by P protein phosphorylation.

**Figure supplement 3—source data 1.** SDS-PAGE showing purified GFP-P$^{WT}$ purified from *E. coli* co-expressing HvCK1.2 or HvCK1.2$^{DN}$ (Related to *Figure 5—figure supplement 3A*).

of GFP-P$^{S5A}$ underwent phase separation like GFP-P$^{WT}$ (*Figure 4—figure supplement 3A*). To examine induction of phase separation by measuring solution turbidity (OD600) as described recently (*Brown et al., 2021*), GFP-P$^{WT}$, GFP-P$^{S5A}$, and GFP proteins (12 µM) were combined with 200 mM NaCl and 20% PEG4000. As expected, the solution turbidity values of the condensed droplets from GFP-P$^{S5A}$ and GFP-P$^{WT}$ were higher than that of free GFP (*Figure 4—figure supplement 3B*). Taken together, these results confirm that phosphorylation of the SR region within the middle IDR of BYSMV P significantly impairs phase separation in vivo and in vitro.

## Host CK1 negatively regulates BYSMV P phase separation

Given the conserved CK1 kinases among host plants and insect vectors that directly target the five Ser residues of the SR motif in vivo and in vitro (*Gao et al., 2020*), it would be interesting to determine whether CK1 affects BYSMV P phase separation in vivo. To this end, GFP-P$^{WT}$, GFP-P$^{S5A}$, and GFP-P$^{S5D}$ were expressed with the empty vector (EV) or the CK1 proteins (NbCK1.3) in *N. benthamiana* leaves. As expected, most of the GFP-P$^{WT}$ granules were dispersed upon co-expression of NbCK1.3, while GFP-P$^{S5A}$ granules were not obviously affected by NbCK1.3 (*Figure 5A* and *Figure 5—figure*

*supplement 1*). Again, GFP-P^S5D was defective in granule formation in either the presence or absence of NbCK1.3 (*Figure 5A*). Western blotting analyses showed that co-expression of NbCK1.3 drastically decreased the hypophosphorylated P72 form of GFP-P^WT compared with equal accumulation of P72 and P74 forms during co-expression of GFP-P^WT and EV (*Figure 5B*, compare lanes 2 and 3). In contrast, the P72 form of GFP-P^S5A and the P74 form of GFP-P^S5D were not affected by co-expression of either NbCK1.3 or EV (*Figure 5B*). These results indicate that host NbCK1 inhibits BYSMV P phase separation mainly by phosphorylating the five Ser residues of the SR motif within the middle IDR region of BYSMV P (*Figure 4A*).

To further determine the effect of NbCK1 on phase separation of GFP-P^WT, we used a loss-of-function mutant (K38R and D128N, NbCK1.3^DN) that has been described previously (*Gao et al., 2020*). As expected, overexpression of NbCK1.3^DN did not affect phosphorylation and phase separation of GFP-P^WT compared with NbCK1.3 (*Figure 5—figure supplement 2*). Since eukaryotic-type protein kinases are absent in *E. coli*, we used a bacterial co-expression system to isolate phosphorylated BYSMV P protein from *E. coli*. We then co-expressed GFP-P^WT with NbCK1.3 or NbCK1.3^DN. Interestingly, the purified GFP-P^WT protein underwent phase separation when co-expressed with NbCK1.3^DN, but co-expression of NbCK1.3 inhibited phase separation of GFP-P^WT (*Figure 5C*). In addition, the SDS-PAGE gel showed that co-expression of NbCK1.3, rather than NbCK1.3^DN, resulted in production of the upper P74 band of GFP-P^WT (*Figure 5D*). Moreover, we failed to detect the P74 band of GFP-P^WT after $\lambda$-protein phosphatase ($\lambda$ PPase) treatment in vitro (*Figure 5E*), indicating that P74 represents hyper-phosphorylated forms of GFP-P^WT elicited by co-expressed NbCK1.3. In agreement with NbCK1, the CK1 orthologue of barley plants (HvCK1.2) suppressed phase separation of GFP-P^WT in co-expression assays (*Figure 5—figure supplement 3*).

Collectively, host CK1 proteins inhibit phase separation of BYSMV P by phosphorylating the SR region of the BYSMV P middle IDR.

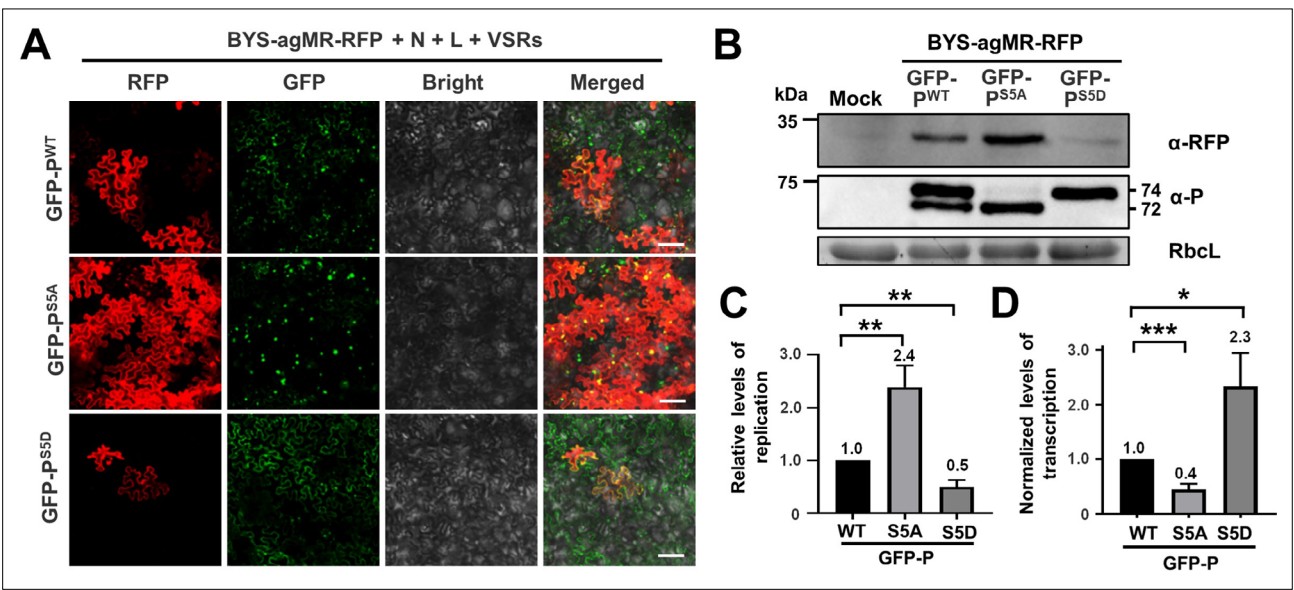

**Figure 6.** Phase separation of GFP-P modulates virus replication and transcription. (**A**) RFP foci in *Nicotiana benthamiana* leaves infiltrated with *Agrobacterium* for co-expression of BYS-agMR-RFP, N, L, and VSRs with GFP-P^WT, GFP-P^S5A, or GFP-P^S5D at 8 days post infiltration (dpi). Scale bars, 100 μm. (**B**) Western blotting analyzing accumulation of RFP, GFP-P^WT, GFP-P^S5A, and GFP-P^S5D in the leaf samples of panel A. (**C**) Quantitative real-time PCR analyzing the relative levels of minigenome replication supported by the GFP-P^WT, GFP-P^S5A, and GFP-P^S5D proteins. (**D**) Quantitative real-time PCR analyzing the relative levels of RFP mRNA in the same samples of A. In panels C and D, error bars indicate SD (n = 3). *p < 0.05, **p < 0.01, and ***p < 0.001 (Student's t-test).

The online version of this article includes the following source data and figure supplement(s) for figure 6:

**Source data 1.** Source data and images of *Figure 6* (Related to *Figure 6B-D*).

**Figure supplement 1.** Illustration of BYS-agMR-RFP infection in epidermal cells of *Nicotiana benthamiana* leaves.

## Condensed phase of BYSMV P facilitates virus replication

We next investigated the relevance of BYSMV P phase separation on replication and transcription. Recently, we have developed a BYSMV minireplicon (BYS-agMR) to mimic viral replication and transcription processes (*Fang et al., 2019*). Based on the pBYS-agMR plasmid, we generated a frame-shift vector (pBYS-agMR-RFP) to abolish translation of the GFP mRNA, which allowed us to observe phase separation of GFP-P during virus replication (*Figure 6—figure supplement 1A*). Then, we used GFP-P$^{WT}$, GFP-P$^{S5A}$, or GFP-P$^{S5D}$ to rescue BYS-agMR-RFP in *N. benthamiana* leaves after co-agroinfiltration of pBYS-agMR-RFP, pGD-VSRs, pGD-N, and pGD-L (*Figure 6—figure supplement 1B*). Consistent with the result above (*Figure 4B*), GFP-P$^{S5A}$ formed larger spherical granules than those of GFP-P$^{WT}$, whereas GFP-P$^{S5D}$ was diffuse in the cytoplasm (*Figure 6A*). Furthermore, the numbers of RFP foci in the GFP-P$^{S5A}$ samples were significantly higher than those of GFP-P$^{WT}$, whereas GFP-P$^{S5D}$ expression resulted in a reduced number of RFP foci (*Figure 6A*). Western blotting analyses consistently showed that RFP accumulation was highest after GFP-P$^{S5A}$ expression, followed by GFP-P$^{WT}$ and GFP-P$^{S5D}$ (*Figure 6B*).

We subsequently performed quantitative reverse transcription PCR (RT-qPCR) to compare relative accumulation of RNA products of virus replication and transcription. As shown in *Figure 6—figure supplement 1B*, the agMR is transcribed from 35S promoter in vivo, and then replicate and produce genomic MR (gMR), accumulation of which represents MR replication level. Based on the gMR as templates, RFP mRNA was transcribed from the intergenic region of gMR. Therefore, the transcriptional activities of these mutants were compared through normalization of RFP mRNA levels relative to the gMR templates (*Figure 6—figure supplement 1B*). Accumulation of the full-length gMR was upregulated in GFP-P$^{S5A}$ samples but decreased in GFP-P$^{S5D}$ samples compared with those of GFP-P$^{WT}$ samples (*Figure 6C*). However, GFP-P$^{S5D}$ supported higher transcription activity but GFP-P$^{S5A}$ expression compromised transcription (*Figure 6D*). Collectively, GFP-P$^{S5A}$ with increased LLPS activity supports enhanced virus replication but decreased transcription. In contrast, GFP-P$^{S5D}$ with impaired LLPS activity inhibited virus replication but facilitated transcription. These results of GFP-tagged proteins are in agreement with our previous studies using the free P$^{WT}$, P$^{S5A}$, and P$^{S5D}$ (*Gao et al., 2020*), suggesting that the GFP tag can indicate phase separation of P but has no effects on replication or transcription of minigenome.

In summary, the unphosphorylated BYSMV-P protein undergoes LLPS and forms spherical granules as viral factories to promote virus replication. In contrast, the conserved CK1 protein kinases phosphorylate the SR region of BYSMV P to hyper-phosphorylated P forms and compromise LLPS, which results in soluble RNPs for virus transcription (*Figure 7*). Therefore, the CK1-mediated phosphorylation inhibits phase separation of BYSMV P and viral replication.

## Discussion

Membraneless virus IBs or viroplasms are hallmarks of NSR virus infections. In recent studies, LLPS has emerged as a critical mechanism in formation of replication factories of animal NSR viruses (*Brocca et al., 2020*; *Su et al., 2021*). In contrast, the mechanisms whereby plant NSR viruses form viroplasms as replication sites have not been characterized. Here, we provide the first evidence, to the best of our knowledge, that a plant NSR virus uses LLPS to form viral replication center. We have identified a novel *Cytorhabdovirus* P protein function that provides a scaffold protein for viroplasm formation. The BYSMV P protein can undergo phase separation alone in vivo and in vitro to concentrate membraneless compartments for virus replication. The NBs of rabies virus were formed through phase separation, which however requires co-expression of both the N and P proteins and residues 132–150 of an intrinsically disordered domain within the P protein (*Nikolic et al., 2017*).

Another intriguing finding in our study is that host CK1-dependent phosphorylation of the BYSMV P protein inhibits phase separation. Our previous study has shown that the conserved CK1 kinases of host plants and insect vectors are responsible for phosphorylation of a SR motif at the middle IDR of BYSMV P (*Gao et al., 2020*). In the present study, we found that the BYSMV P granules were dispersed by phosphorylation-mimic mutations (P$^{S5D}$) of the SR motif or overexpression of CK1 proteins in vivo (*Figures 4B and 5A*). These results are in agreement with recent reports in mammalian cells in which CK2-mediated phosphorylation of zona occludens, a cytoplasmic scaffolding protein, inhibits homologous phase separation (*Beutel et al., 2019*). The partitioning defective 3/6 (Par3/Par6)

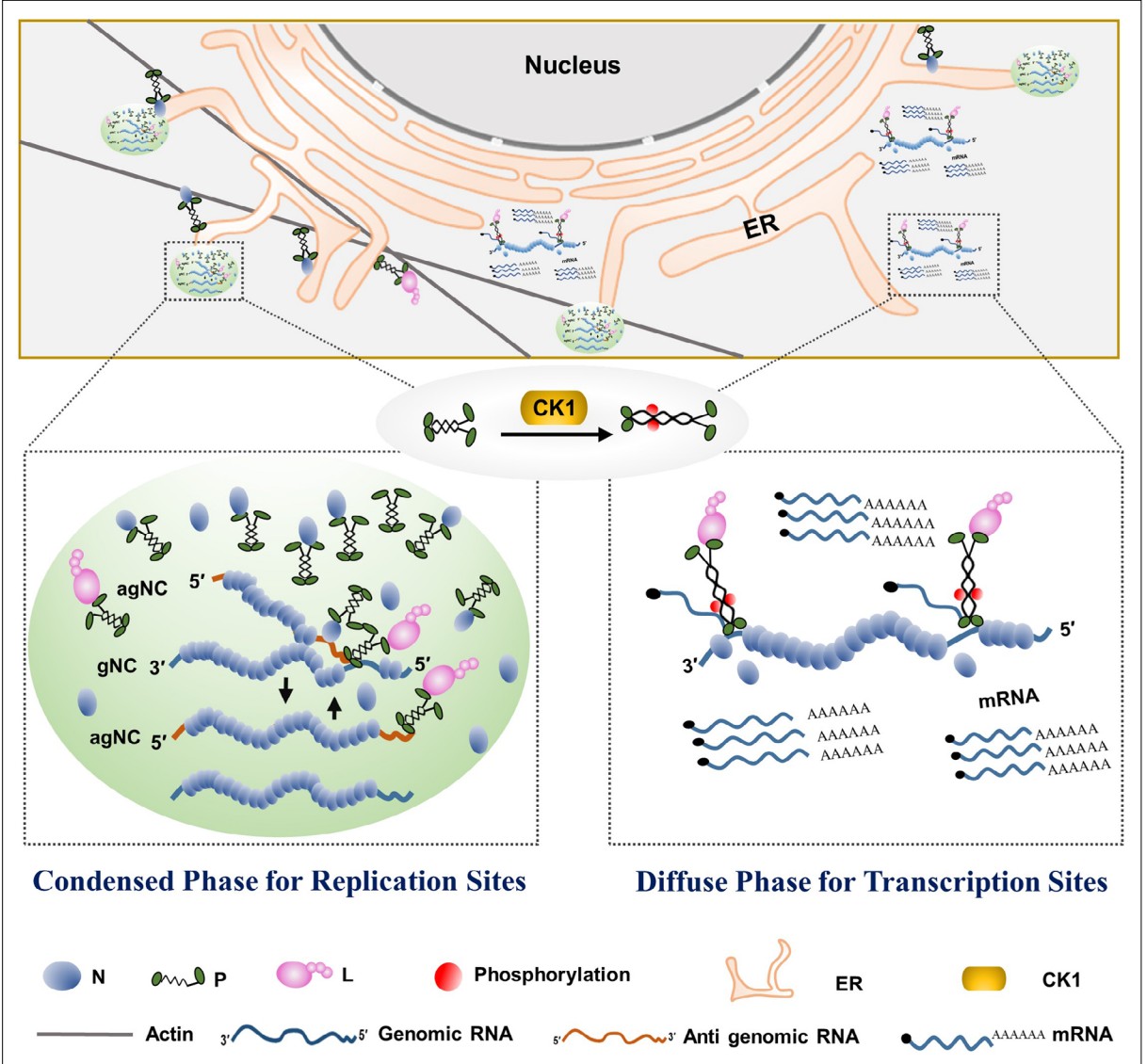

**Figure 7.** Model for phase separation of barley yellow striate mosaic virus (BYSMV) phosphoprotein (P) in modulating rhabdovirus replication and transcription. Rhabdovirus replication requires high concentration of viral N protein for encapsidating newly synthesized genomic/antigenomic RNA. Thus, unphosphorylated BYSMV P undergoes phase separation, and then recruits the nucleotide (N) and polymerase (L) proteins, as well as genomic RNAs into membraneless condensates for optimal replication. In addition, the granules move along the ER/actin network and fuse with each other. In the transcription sites, the serine-rich (SR) motif in the middle intrinsically disordered region (IDR) of BYSMV P is phosphorylated by casein kinase 1 (CK1), and the resulting hyper-phosphorylated P is unable to undergo phase separation, which facilitates virus transcription and viral mRNA release for viral protein translation.

condensates are also dispersed by PKC-mediated phosphorylation (*Liu et al., 2020*). Thus, precisely balanced phosphorylation states may represent additional regulators of protein phase separation and cellular functions.

Rhabdovirus transcription and replication complexes contain common virus replication derivatives, in which the N-RNA complex serves as a template for viral RdRp complexes comprising the P and L proteins (*Ivanov et al., 2011*). Therefore, how to uncouple rhabdovirus transcription and replication processes has remained elusive for many years because both complexes contain the N, P, and L proteins and the viral genomic RNA. Here, we demonstrate that the phosphorylation mutants ($P^{S5A}$) in the SR motif of BYSMV P have high LLPS activity that is coupled with enhanced virus replication. On the other hand, the phosphorylation-mimic mutant ($P^{S5D}$) is impaired in LLPS, which facilitates virus transcription but significantly inhibits virus replication. Therefore, these results suggest that

condensed phase of RNPs as viroplasms are a positive regulator for virus replication, but inhibit virus transcription. During rhabdovirus replication, the newly synthesized genome or antigenome need to be encapsidated by the N protein, which requires a continuous supply of RNA-free N molecules (*Ivanov et al., 2011*). Therefore, it is not surprising to find that LLPS of BYSMV P facilitates concentration of the BYSMV N protein into virus replication sites. In contrast, viral mRNAs are released without N encapsidation, which results in dispersed protein rather than condensed N protein in virus transcription sites (*Figure 7*). Therefore, we propose that the phosphorylation of the BYSMV P SR region by host CK1s abrogates the liquid-like membraneless viroplasms and subsequently drives the switch from virus replication to transcription.

Emerging evidence shows that the biogenesis and dynamics of phase-separated membraneless condensates require membranes as assembly platforms for transportation (*Zhao and Zhang, 2020*). For instance, ER contact sites regulate the biogenesis and fission of two important membraneless organelles, processing bodies (PBs) and liquid spherical stress granules (*Lee et al., 2020*). More interestingly, mRNA translation inhibition facilitates formation of PB condensates, whereas increasing the translational activity induces PB disassembly, indicating that the ER contact sites may shuttle mRNAs between the ER and PBs (*Lee et al., 2020*). In the current study, we observed that the ER/actin network acts as a platform to facilitate trafficking and fusion of the GFP-P granules (*Figure 1G*; *Figure 1—video 1*). Given that viroplasms need large amount of BYSMV N to encapsidate newly synthesized genomic/antigenomic RNAs, the BYSMV P-mediated condensates on the ER/actin network opens up the possibility that ER-viroplasm contact sites are conduits for viral protein and mRNA exchange between the two organelles. Taken together, we propose that the BYSMV P condensates tether on the ER tubules and are dispersed by CK1-mediated phosphorylation. Subsequently, virus transcription is stimulated and viral mRNAs are transported to the ER for efficient translation.

Although the roles of LLPS in viral infections are emerging, multilayered regulatory mechanisms controlling assembly and disassembly of protein condensates remain to be explored. Here, we provide evidence that a plant *Cytorhabdovirus* P protein-mediated LLPS triggers formation of localized viral protein condensates for optimized virus replication. Currently, we cannot assemble active viroplasms in vitro, because the larger L polymerase protein (234 kDa) is difficult to express and purify from *E. coli*. Therefore, we cannot directly measure the contribution of two P phosphorylation states ($P^{S5A}$ and $P^{S5D}$) to virus replication and transcription in vitro. Furthermore, future studies are required to identify the spatiotemporal control mechanisms of host factors including CK1 in viral protein phase separations. Another interesting but unresolved question is how cellular membranes, such as the plasma membrane or ER, regulate the biogenesis and dynamics of viroplasm condensates during virus infection. Nonetheless, these results increase our understanding of the distinct roles of host CK1 and LLPS in rhabdovirus transcription and replication.

# Materials and methods

**Key resources table**

| Reagent type (species) or resource | Designation | Source or reference | Identifiers | Additional information |
|---|---|---|---|---|
| Strain, strain background (*Escherichia coli*) | BL21 | Thermo Fisher | Cat#C600003 | |
| Strain, strain background (*Agrobacterium tumefaciens*) | EHA105 | Weidibiotechnology | Cat#AC1012 | |
| Antibody | Anti-GST (Rabbit polyclonal) | Abcam | Cat#ab9085 | WB(1:5000) |
| Antibody | Anti-FLAG (Mouse monoclonal) | Sigma | Cat#F1804 | WB(1:5000) |
| Antibody | Anti-RFP (Rabbit polyclonal) | *Fang et al., 2019* | | WB(1:3000) |
| Antibody | Anti-BYSMV-P (Rabbit polyclonal) | *Fang et al., 2019* | | WB(1:3000) IEM(1:500) |
| Antibody | Anti-Mouse IgG (H + L)-HRP Conjugate (Goat polyclonal) | Bio-Rad | Cat#170–6516 | WB(1:20,000) |

*Continued on next page*

*Continued*

| Reagent type (species) or resource | Designation | Source or reference | Identifiers | Additional information |
|---|---|---|---|---|
| Antibody | Anti-Rabbit IgG (H + L)-HRP Conjugate (Goat polyclonal) | EASYBIO | Cat#BE0101 | WB(1:20,000) |
| Antibody | Anti-Rabbit IgG-gold Conjugate (Goat polyclonal) | Sigma | Cat#G7402 | IEM(1:50) |
| Commercial assay or kit | RiboMAXTM Large Scale RNA Production System-T7 | Promega | Cat#P1300 | |
| Chemical compound, drug | Latrunculin B | Abcam | Cat#ab144291 | |
| Software, algorithm | GraphPad Prism 8 | PMID:22434839 | RRID:SCR_002798 | |
| Software, algorithm | ImageJ | PMID:22930834 | RRID:SCR_003070 | |

## Plasmid construction

Based on the pBYSMV-agMR vector (*Fang et al., 2019*), the inverse PCR product containing an adenine insertion after the start codon of RsGFP was self-ligated to generate the pBY-agMR-RFP vector. To engineer the pGD-GFP-P$^{WT}$, pGD-GFP-P$^{S5A}$, and pGD-GFP-P$^{S5D}$ constructs, the P$^{WT}$, P$^{S5A}$, and P$^{S5D}$ ORFs were amplified from pBYG-P$^{WT}$, pBYG-P$^{S5A}$, and pBYG-P$^{S5D}$ (*Gao et al., 2020*), respectively, and then introduced into the pGDG vector (*Goodin et al., 2002*).

The pGD-spGFP1-10-P and pGD-4× spGFP11P vectors were obtained by replacing the GFP sequence of pGD-GFP-P$^{WT}$ with the cDNA sequences of spGFP1-10 and 4× spGFP11 (*Pédelacq et al., 2006*; *Pedelacq and Cabantous, 2019*), respectively. To generate pGD-4× spGFP11-UBC32, the P sequence of pGD-4× spGFP11-P was replaced with the cDNA sequence corresponding to the N-terminal 64 amino acids of UBC32 (AT3G17000) as described previously (*Li et al., 2020*). The LRR84A-GS-2× spGFP11 plasmid has been described previously (*Li et al., 2020*).

To obtain pET-30a-GFP, pET-30a-GFP-P$^{WT}$, pET-30a-GFP-P$^{S5A}$, pET-30a-GFP-P$^{S5D}$ for recombinant protein expression, the cDNA sequences of 6× His GFP, 6× His-GFP-P$^{WT}$, 6× His-GFP-P$^{S5A}$, or 6× His-GFP-P$^{S5D}$ were amplified and then inserted into the pET-30a vector for expression of 6× His-tagged proteins. The *mCherry* sequence was cloned into pET-32a vector (Novagen) to generate the pET-32a-mCherry vector, in which the BYSMV N ORF was inserted to generate the pET-32a-mCherry-N expression vector.

The pBYSMV-agMR, pGD-N, pGD-P, pGD-L, pGD-VSRs, pGD-ECFP-N, pSuper-mCherry-L, pGDG-NCMV P$^{WT}$, mCherry-HDEL, and GFP-ABD2-GFP plasmids have been described previously (*Fang et al., 2019*). The constructs of pMDC32-NbCK1.3, pGEX-NbCK1.3, pGEX-NbCK1.3$^{DN}$, pGEX-HvCK1.2, and pGEX-HvCK1.2$^{DN}$ have been described previously (*Gao et al., 2020*), as has pCNX-mRFP (*Li et al., 2020*).

All sequences were amplified using 2× Phanta Max Master Mix (Vazyme Biotech Co., Ltd) and inserted into vectors using a ClonExpress Ultra One Step Cloning Kit (Vazyme Biotech Co., Ltd). Sanger sequencing was performed to confirm sequences. The primers used in this study are listed in *Supplementary file 1*.

## Plant materials and protein expression in vivo

Four-week-old *N. benthamiana* plants were used for agroinfiltration. The plants were grown in a growth chamber with 16/8 hr light/dark cycles and 24°C/20°C (day/night) temperatures. Agroinfiltration experiments for transiently expressing proteins were performed as described previously (*Fang et al., 2022*). For subcellular localization experiments, *Agrobacterium* harboring plasmids expressing pGD-GFP-P$^{WT}$, pGD-GFP-P$^{S5A}$, pGD-GFP-P$^{S5D}$, pGD-CFP-N, pSuper-L-mCherry, pGD-GFP, pGD-CFP, pSuper-mCherry, pGDG-NCMV P (OD600, 0.3), pGD-spGFP$_{1-10}$-P (OD600, 0.3), pGD-4×spGFP$_{11}$-UBC32 (OD600, 0.5), LRR84A-GS-2× spGFP11 (OD600, 0.2), HDEL-mCherry, CNX-mRFP, pMDC32-NbCK1.3 (OD600, 0.1), or pMDC32-NbCK1.3$^{DN}$ (OD600, 0.6) were mixed with TBSV P19 (OD600, 0.1) for infiltration assays. For BYSMV MR assays, *Agrobacterium* harboring pBYS-agMR-RFP, pGD-N, pGD-P/GFP-P$^{WT}$/GFP-P$^{S5A}$/GFP-P$^{S5D}$, pGD-L, pGD-VSRs were diluted to OD600 of 0.3, 0.1, 0.1, 0.3, and 0.1, respectively.

## Expression and purification of recombinant proteins

All recombinant plasmids were transformed into *E. coli* BL21 (DE3) cells for recombinant protein expression as described previously (*Tong et al., 2021*). Briefly, BL21 cells containing different plasmids were grown in 3 mL LB medium with 100 µg/mL kanamycin overnight at 37°C. The pre-culture was transferred into 1 L LB media with 100 µg/mL kanamycin and grown to OD600 of 0.5 at 37°C, and then induced by 0.5 mM isopropyl β-D-1-thiogalactopyranoside for 18–24 hr at 18°C. Bacterial cells were collected and suspended in lysis buffer (30 mM Tris-HCl pH 7.5, 500 mM NaCl, 1 mM PMSF, and 20 mM imidazole) before being sonicated. After centrifugation (39,000× $g$, 60 min), the supernatant was flowed through a column containing 2 mL Ni–nitrilotriacetic acid resin equilibrated with lysis buffer. After washing in buffer (30 mM Tris-HCl pH 7.5, 500 mM NaCl, and 40 mM imidazole), recombinant proteins were eluted with elution buffer (30 mM Tris-HCl pH 7.5, 500 mM NaCl, and 400 mM imidazole). For kinase assays in *E. coli*, plasmids encoding 6× His-GFP-P$^{WT}$ with pGEX-NbCK1.3, pGEX-NbCK1.3$^{DN}$, pGEX-HvCK1.2, or pGEX-HvCK1.2$^{DN}$ were co-transformed into *E. coli* Rosetta cells. Expression and purification of 6× His-GFP-P$^{WT}$ were performed as described above (*Gao et al., 2020*).

## In vitro liquid droplet reconstitution assays

All recombinant proteins were centrifuged at 15,000× $g$ for 10 min to remove aggregates. The protein concentrations were determined with a NanoDrop spectrophotometer (NanoDrop Technologies) before phase separation assays. All proteins were diluted with buffer (30 mM Tris-HCl pH 7.5, 1 mM DTT) to desired protein and salt concentrations. Unless indicated, the final concentration of NaCl was 125 mM and all experiments were performed at room temperature. Phase separation between GFP-P and mCherry-N was conducted by mixing GFP-P with mCherry-N to final concentrations of 10 and 6 µM, respectively.

For droplet assembly and turbidity assay, GFP, GFP-P$^{WT}$, or GFP-P$^{S5A}$ (final concentration, 12 µM) were incubated within buffer containing 20% PEG4000, 200 mM NaCl, 30 mM Tris-HCl (pH 7.5), and 1 mM DTT. Samples were incubated at room temperature for 5 min, and the OD600 values of 60 µL samples were measured using SpectraMax i3xm.

For Cy5-labeled RNAs, 5–10 µg DNA templates of T7 promoter-driven BYSMV trailer sequence served as templates for in vitro transcription with the RiboMAX Large Scale RNA Production Systems (Promega, P1300) based on the manufacturer's protocols. Note that the final concentration of ATP, CTP, GTP, UTP, and Cy5-UTP (ApexBio, B8333) in the mixtures were 1.75, 1.75, 1.75, 0.875, and 0.175 mM, respectively. The reactants were mixed gently and incubated at 37°C for 3.5 hr, followed by addition of RNase-Free DNase I for 15 min to remove DNA templates. Then, 0.1 volume of 3 M sodium acetate (pH 5.2) and 2.5 volumes of 100% ethanol mixtures were added, followed by storage at –20°C for more than 4 hr. After centrifugation, the precipitated RNA was washed with 75% ethanol, and suspended in nuclease-free water, and heated at 95°C for 5 min. Phase separation of GFP-P, mCherry-N, and Cy5-labeled RNAs was carried out by mixing the indicated proteins and Cy5-labeled RNAs, and diluting with buffer (30 mM Tris-HCl pH7.5, 1 mM DTT) to the desired concentrations.

## Confocal laser scanning microscopy and image processing

*N. benthamiana* leaves were agroinfiltrated with *Agrobacterium tumefaciens* containing various constructs and subjected to live-cell imaging at 2–3 dpi with a Leica TCS-SP8 laser scanning confocal microscope. For BYSMV infectivity assays, *N. benthamiana* leaves were observed at 6 days after infiltration with BYS-agMR-RFP. CFP, GFP, mCherry/RFP, and Cy5 were excited at 440, 488, 568, or 633 nm, and detected at 450–490, 500–540, 585–625, or 638–759 nm, respectively. Time-series programs were used to obtain videos. For each video, more than 50 consecutive images were taken at 3–5 s intervals (in vivo) or 20 s intervals (in vitro), and six images per second were edited using the Fiji/ImageJ software. Unless indicated, all images were processed using Leica SP8 software.

Image processing of granule numbers and sizes was carried out as described previously with minor modification (*Brown et al., 2021*). The GFP-P$^{WT}$, GFP-P$^{S5A}$, and GFP-P$^{S5D}$ proteins were individually expressed in *N. benthamiana* leaves by agroinfiltration. Images were captured at 3 dpi. All images were captured under the same parameters with a field of 175 µm × 175 µm, and 8–10 representative fields were captured from more than five leaves. Then, raw images were imported into the ImageJ software (*Schneider et al., 2012*), converted to grayscale (8-bit), and adjusted the threshold

to 50–255. The numbers and sizes of granules were analyzed using the 'analyze particles' function and imported into Excel tables. The numbers of granules (>0.2 µm$^2$) were counted and analyzed.

## Transmission electron and immunoelectron microscopy

TEM assays were performed as described previously (*Yan et al., 2015*). Briefly, stems of mock-treated or rBYSMV-RFP-infected barley plants were fixed and embedded in Spurr's resin, and ultra-thin sections were observed with a Hitachi TEM system. Immunogold labeling was performed as described methods with minor modifications (*Jin et al., 2018*). Briefly, stems of healthy or rBYSMV-RFP-infected barley plants were incubated within a mixture of 3% formaldehyde, 4% Suc, 0.1% glutaraldehyde in 0.1 M phosphate buffer (pH 7.2), treated with vacuum infiltration, and fixed at 4°C for 2 hr. The stems were dehydrated in 30%, 50%, 70%, 80%, 95%, and 100% of ethanol, and then incubated in increasing concentrations of 50%, 75%, and 100% Lowicryl K4M resin for polymerization under 360 nm UV light at –20°C for 3 days and then at 25°C for 2 days. After polymerization, blocks were cut into ultra-thin sections that were collected on Formvar-coated nickel grids. To reduce nonspecific binding, the grids were incubated in 0.01 M PBS (pH 7.2) for 30 min and then blocked in 3% BSA (dissolved in PBS) for 15 min at 25°C. Then, the grids were incubated with primary rabbit polyclonal anti-P (1:500) antibodies overnight at 4°C. After washing three times with 0.01 M PBS (pH 7.2) buffer, the grids were incubated with goat anti-rabbit secondary antibodies conjugated with 10 nm gold particles (Sigma, Cat#G7402), followed by rinsing with 0.01 M PBS (pH 7.2) buffer for two times and ddH$_2$O for two times. Finally, sections were stained with uranyl acetate and Reynolds' lead citrate prior to viewing with a Hitachi TEM system.

## Subcellular localization in barley and maize protoplasts

Isolation barley (Golden Promise) and maize (inbred line Zheng158) protoplasts infected by mock buffer or rBYSMV-RFP (15 dpi) and polyethylene glycol (PEG)-mediated transfection were conducted as described previously (*Zhu et al., 2014*). Approximately 10 µg pGD-GFP or pGD-GFP-P plasmids were gently mixed with 100 µL of protoplasts (1 × 10$^5$) and transfected by PEG4000. The transfected protoplasts were harvested 16–18 hr post transfection for fluorescence detection.

## Fluorescence recovery after photobleaching

FRAP were performed with a Leica SP8 laser scanning confocal microscope (63×/100× oil objective, PMT detector). *N. benthamiana* leaves were agroinfiltrated to express the GFP-P$^{WT}$ protein, and then subjected to living-cell imaging at 2–3 dpi. Note that *N. benthamiana* leaves were treated with 10 mM LatB (Abcam) to inhibit trafficking of GFP-P$^{WT}$ granules at 3 hr before FRAP assays. Then the GFP-P granules were bleached three times with a 488 nm laser at 100% laser power and time-lapse modes were used to collect recovery images. For in vitro FRAP assays, droplets were bleached once with a 488 nm laser at 50% laser power with ≥12 samples. The FRAP data analysis were conducted as described previously (*Boeynaems et al., 2017*). The recovery curves were carried out with GraphPad Prism8 software.

## Western blotting analysis

Total proteins were isolated from *N. benthamiana* leaf tissues in extraction buffer (100 mM Tris-HCl, pH 6.8, 20% glycerol, 4% SDS, 0.2% bromphenol blue, 5% β-mercaptoethanol). Total proteins were separated in a 4–15% SDS-PAGE gradient and transferred to nitrocellulose membrane (GE Healthcare Life Sciences). Membranes were blocked with 5% (m/v) skimmed milk powder at room temperature for 1 hr and then incubated with primary antibodies at 37°C for 1 hr. After washing three times, membranes were incubated with secondary antibodies at 37°C for 1 hr. Antibodies against RFP (1:3000), P (1:3000), GST (1:5000), GFP (1:5000; MBL, 598), Flag (1:5, 000; Sigma, F1804) were used for protein detection. Goat anti-rabbit IgG (EASYBIO, BE0101) and goat anti-mouse IgG horseradish peroxidase conjugate (Bio-Rad, 170–6516) were used as secondary antibodies. After addition of NcmECL Ultra stabilized peroxide reagent (NCM Biotech, P10300B), chemiluminescence of membranes was detected with a Biomolecular Imager (Azure biosystems, Inc).

## BYSMV P granule sedimentation assay

Granule purification assays were performed as described previously (*Hubstenberger et al., 2017*). *N. benthamiana* leaves (0.3 g) expressing GFP-P$^{WT}$, GFP-P$^{S5A}$, and GFP-P$^{S5D}$ were grounded in liquid nitrogen and suspended in 600 µL lysis buffer (50 mM Tris-HCl, pH 7.5, 1 mM EDTA, 150 mM NaCl, 1 mM DTT, 0.2% Triton X-100). After centrifuging at 13,000× *g* for 10 min, the supernatants were added 600 µL extraction buffer (100 mM Tris-HCl, pH 6.8, 20% glycerol, 4% SDS, 0.2% bromphenol blue, 5% β-mercaptoethanol) and used as soluble protein samples. The pellets were suspended in 1 mL lysis buffer and centrifuged at 6000× *g* for 10 min to deplete free GFP-P$^{WT}$, GFP-P$^{S5A}$, and GFP-P$^{S5D}$, and the pellets were resuspended in 200 µL extraction buffer for use as pellet samples. For input samples, 0.1 g of grounded *N. benthamiana* leaves were suspended in 500 µL extraction buffer. All the input, supernatant, and pellet samples were used to detect GFP-P accumulation by Western blotting analyses with antibodies against BYSMV P.

## In vitro dephosphorylation assays

The phosphorylated GFP-P$^{WT}$ protein, purified from *E. coli* Rosetta cell containing pGEX-NbCK1.3 or pGEX-HvCK1.2 plasmid, was incubated with 10 U/µL Lambda Protein Phosphatase (New England Biolabs, #P0753) at 30°C for 30 min. Samples were resuspended in extraction buffer (100 mM Tris-HCl, pH 6.8, 20% glycerol, 4% SDS, 0.2% bromphenol blue, 5% β-mercaptoethanol) and analyzed by SDS-PAGE.

## RT-qPCR assays

The RT-qPCR assay was performed as described previously (*Gao et al., 2020*). Briefly, total RNA isolated from plants was first treated with DNase I (Takara) to remove DNA contamination. Next, 2.5 µg total RNA was used as a template for reverse transcription by HiScript II Reverse Transcriptase (Vazyme Biotech Co., Ltd) with primers oligo (dT)/BYS-RT-F and qNbEF1α-R. Quantitative PCR (qPCR) was performed using SsoFast EvaGreen Supermix (Bio-Rad) on CFX96 Real-Time system (Bio-Rad). The *EFIA* gene was used as an endogenous control. Three independent biological replicates were used for biological statistics analysis. All the primers used in this study are provided in *Supplementary file 1*.

## Prediction of IDRs

The IDRs were predicted with the online tool PONDR (http://www.pondr.com/) with default parameters.

## Quantification and statistical analyses

Images were analyzed with Fiji/ImageJ software. At least three independent replicates were used for all experiments, and statistical analyses were done using the GraphPad Prism8 software. Statistical significance was assessed by unpaired two-tailed Student's t-test.

# Acknowledgements

We thank Prof. Jialin Yu, Chenggui Han, and Yongliang Zhang for their helpful discussion, and Yan Zhang for her technique assistance. We thank Prof. Caiji Gao (South China Normal University) for providing the CNX-mRFP, spGFP1-10-UBC32, and LRR84A-GS-2× spGFP11 plasmids. We thank Prof. Xiaorong Tao (Nanjing Agricultural University) for his gift of the GFP-ABD2-GFP and HDEL-mCherry plasmids. This work was supported by the Natural Science Foundation of China 31872920 (XBW) and 32102150 (QG), as well as China Postdoctoral Science Foundation 2021T140713 (QG).

# Additional information

### Funding

| Funder | Grant reference number | Author |
| --- | --- | --- |
| National Natural Science Foundation of China | 31872920 | Xian-Bing Wang |

| Funder | Grant reference number | Author |
|---|---|---|
| National Natural Science Foundation of China | 32102150 | Qiang Gao |
| China Postdoctoral Science Foundation | 2021T140713 | Qiang Gao |

The funders had no role in study design, data collection and interpretation, or the decision to submit the work for publication.

## Author contributions

Xiao-Dong Fang, Conceptualization, Formal analysis, Investigation, Methodology, Validation, Visualization, Writing - original draft; Qiang Gao, Investigation, Methodology, Resources, Validation; Ying Zang, Formal analysis, Methodology, Writing - review and editing; Ji-Hui Qiao, Investigation, Methodology, Validation, Writing - review and editing; Dong-Min Gao, Wen-Ya Xu, Investigation, Methodology, Visualization; Ying Wang, Project administration, Validation, Writing - review and editing; Dawei Li, Investigation, Project administration, Validation; Xian-Bing Wang, Conceptualization, Funding acquisition, Project administration, Resources, Supervision, Writing - original draft, Writing - review and editing

## Author ORCIDs

Dawei Li http://orcid.org/0000-0003-4133-1263
Xian-Bing Wang http://orcid.org/0000-0003-3082-2462

## Decision letter and Author response

Decision letter https://doi.org/10.7554/eLife.74884.sa1
Author response https://doi.org/10.7554/eLife.74884.sa2

# Additional files

## Supplementary files

- Supplementary file 1. Primers used in this study.
- Transparent reporting form

## Data availability

All data generated or analysed during this study are included in the manuscript and supporting file.

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
