## [Editor Report]

This paper reveals that the phosphoprotein (P) of a plant negative-sense RNA virus, Barley yellow striate mosaic virus (BYSMV), forms condensates through liquid-liquid phase separations (LLPSs). Using BYSMV minireplicon system, they show that the unphosphorylated P protein undergoes phase separation to promote virus replication. Whereas, the host casein kinase 1 (CK1) phosphorylates P protein to inhibit phase separation and virus replication.

---

## [Decision Letter]

**Decision letter after peer review:**

Thank you for submitting your article "Host casein kinase 1-mediated phosphorylation modulates phase separation of a rhabdovirus phosphoprotein and virus infection" for consideration by *eLife*. Your article has been reviewed by 3 peer reviewers, one of whom is a member of our Board of Reviewing Editors, and the evaluation has been overseen by Detlef Weigel as the Senior Editor. The following individual involved in review of your submission has agreed to reveal their identity: Xiaoming Zhang (Reviewer #3).

Essential revisions:

1) Specificity of 1,6-hexanediol treatment on BYSMV infection should be demonstrated by including a control virus infection.

2) To claim that phase separation of P protein modulates replication vs transcription, NbCK1 should be knockdown/knockout and assess the effect on P protein phase separation and on viral transcription and replication.

3) Phase separation property of P protein should be demonstrated when delivered using full-length infectious clone or mini-replicon in its natural host barley.

4) Include statistical tests and quantification of FRAP and other microscopy data presented in the manuscript with appropriate controls and normalization.

5) Explain the mini-replicon vectors with GFP and RFP and the related experiments to clarify the difference between replication vs transcription so that non-virologists could understand why making a frameshift mutation only abolishes GFP fluorescence but not RFP. Unless additional experimental data is included in the revision regarding mini-replicon experiments, soften the claim that the phase separation of P modulates replication vs. transcription.

6) Either soften the claim that viral genomic RNA is recruited to the P-N droplets or include data showing that the full-length BYSMV RNA is indeed recruited to the droplets.

7) Remove hollow droplets data of P protein or move the data to supplement.

*Reviewer #1 (Recommendations for the authors):*

In summary, this is an interesting paper. Authors should show that P protein produced during normal infection does indeed undergo phase separation. Authors have necessary tools to demonstrate this. Furthermore, they should by some genetic experiments that NbCK1 is the one that regulates this process.

*Reviewer #2 (Recommendations for the authors):*

The authors conclusively demonstrate that P protein phase separates and can be inhibited by CK1 kinase activity. These are the strengths of the study. The conclusions that P phase separation favors replication vs. transcription is not as strongly supported. The minireplicon system data is confusing and the use of 1,6-HD is not sufficient for concluding that phase separation of BYSMV P protein is essential for virus fitness.

Below are issues that should be addressed before publication:

1. The use of 1,6-hexanediol in Figure 6 is not a suitable approach for multiple reasons.

a. 1,6-HD is toxic to cells (see reference in public review).

b. 1,6-HD disrupts kinase and phosphatase activity (see ref above).

c. The authors do not demonstrate that 1,6-HD inhibited P phase separation in the insect vector or barley (i.e. how do you know the treatment worked?).

d. Reduced virus accumulation is likely a result of cell toxicity.

2. FRAP data is not properly normalized.

a. Figure 1D: The authors state that approximately 80% of the signal recovered following FRAP. However, this is misleading when the photobleaching only reduced the signal to 20%. The actual recovery was closer to 60%. See https://bio-protocol.org/e2525 for an example of FRAP normalization. Typically, the post bleach signal should be set at 0% to give a more discernable recovery. This issue is also present in Figure 2E. 60% of the signal did not recover based on the data present. The FRAP data must be properly normalized. The %recovery in Figure 2E is less than 50% but the images presented in Figure 2D show much higher recovery (this reviewer sees >80% recovery, possibly higher).

3. Minireplicon data must be better described (including the experimental design).

a. Most readers will not be familiar with rhabdovirus replication and minireplicon systems. The system needs to be better described.

b. It is unclear how S5D can have the highest RFP transcription, but the lowest RFP translation. Is translational repression occurring? The data in panel D does not match the confocal or western blot data concerning RFP expression.

c. How was replication/transcription differentiated between 2x35S-driven transcription?

*Reviewer #3 (Recommendations for the authors):*

Thus, I recommend this paper to publish in your journal, *eLife*.

In the following, I presented a few concerns in regarding to the conclusion of this paper that the authors may want to consider when revising the manuscript. Major concerns:

1) Although the authors claimed that "BYSMV P protein forms liquid-like granules through LLPS in vivo", they did not show any phase separation results of BYSMV P protein in barley, the real host of BYSMV. Although they showed some granular structure in Figure S1A (the quality of Figure 1A is bad), they didn't provide evidence to prove these granules are generated by BYSMV. Instead of granules generated by viruses, these granules may be host P bodies or stress granules that react to viral infections. The author can perform immune-electron microscope or other experiments to prove that these granules are indeed formed by P protein.

2) In Figure 6, the author revealed that the phosphorylation of P protein modulates the proliferation of viruses. Although the reviewer believes that the phase separation of P protein facilitates the viral proliferations, the author does not provide direct evidence to prove that. They only showed the correlation of P protein phase separation and viral proliferations. As it is very hard to prove this hypothesis in vitro, the author can lower the tone of their statements.

3) By detecting the accumulations of full-length genome minigenome and the mRNA in Figure 6C and 6D, the author claimed that the phase separation of P protein enhances replication but suppress transcription. However, the accumulations of virus transcript may not only be caused by the increased biogenesis, but also may be the results of decreased degradation. Again, the author needs to tone down their conclusion.

---

## [Author Response]

Essential revisions:1) Specificity of 1,6-hexanediol treatment on BYSMV infection should be demonstrated by including a control virus infection.

As suggestion of three Reviewers, 1, 6-hexanediol induced cellular toxicity and hereby affected infections of BYSMV and other viruses nonspecifically. In addition, 1, 6-hexanediol would inhibit LLPS of cellular membraneless organelles, such as P-bodies, stress granules, cajal bodies, and the nucleolus, which might also affect different virus infections directly or indirectly. In Figure 6A, GFP-P^S5A^ underwent phase separation and supported efficient replication of BYSMV. In contrast, GFP-P^S5D^ could not LLPS and supported very less replication of BYSMV, indicating important roles of BYSMV P LLPS in viral replication. Therefore, we removed these results of 6F-J, which does not affect the main conclusion of our results.

2) To claim that phase separation of P protein modulates replication vs transcription, NbCK1 should be knockdown/knockout and assess the effect on P protein phase separation and on viral transcription and replication.

Thanks for your suggestions. Indeed, there are some technical reasons in RNAi and overexpressed in barley plants for the following reasons. In our previous studies, BLAST searches showed that the *N. benthamiana* and barley genomes encode 14 CK1 orthologues, most of which can phosphorylated the SR region of BYSMV P. However, multiple CK1 proteins with low homology sequences could phosphorylate the BYSMV P protein, thus it is difficult to silence many CK1 genes at the same time. Secondly, overexpression of CK1 would induce cell death probably some other targets of CK1 proteins

Accordingly, we generated a point mutant (K38R and D128N) in HvCK1.2, in which the kinase activity was abolished. Overexpression of HvCK1.2^DN^ inhibit endogenous CK1-mediated phosphorylation of BYSMV P, indicating that HvCK1.2^DN^ is a dominant-negative mutant. Overexpression of HvCK1.2^DN^ in BYSMV vector inhibit virus infection by impairing the balance of endogenous CK1-mediated phosphorylation in BYSMV P. These results have been described in our recent studies (Gao et al., 2020, Figure 8 and S14).

Here, overexpression of wild type NbCK1 can abolish LLPS of BYSMV P. In contrast, overexpression of NbCK1^DN^ did not phosphorylate the P protein or inhibit LLPS of BYSMV P. Please see Figure 5—figure supplement 2.

Reference:

Gao Q, et al., 2020. Casein kinase 1 regulates cytorhabdovirus replication and transcription by phosphorylating a phosphoprotein serine-rich motif. The Plant Cell 32(9): 2878-2897.

3) Phase separation property of P protein should be demonstrated when delivered using full-length infectious clone or mini-replicon in its natural host barley.

We agree with the reviewer and appreciate the helpful suggestion. BYSMV is a negative-stranded RNA virus, and its inoculation is strictly dependent on transmission of the small brown planthopper. We have tried to fuse GFP to BYSMV P in the full-length infectious clones. Unfortunately, we could not rescue BYSMV-GFP-P into barley plants through insect transmission.

Nonetheless, we carried out immune-electron microscope using BYSMV-infected barley leaves to show the presence of BYSMV P protein in the viroplasms. Please see Figure 1—figure supplement 1.

In addition, we further showed that GFP-P underwent phase separation in protoplasts of barley and maize leaves, as well as rBYSMV-RFP-infected leaves. Please see Figure 1—figure supplement 5.

4) Include statistical tests and quantification of FRAP and other microscopy data presented in the manuscript with appropriate controls and normalization.

We agree with the reviewer and appreciate the helpful suggestions. As the suggested method of Reviewer 2, we analyzed again the data of FRAP to include statistical tests and quantification. We also added mock or healthy controls and large views of confocal results. Please see Figure 1A, 2E, and Supplement Figures.

5) Explain the mini-replicon vectors with GFP and RFP and the related experiments to clarify the difference between replication vs transcription so that non-virologists could understand why making a frameshift mutation only abolishes GFP fluorescence but not RFP. Unless additional experimental data is included in the revision regarding mini-replicon experiments, soften the claim that the phase separation of P modulates replication vs. transcription.

Thank you for your questions and we are sorry for the lack of clarity regarding to the mini-replicon vectors. In our previous studies (Fang et al., 2019), Co-expression of agMR, N, P, L, and VSRs is essential for GFP and RFP protein expression, indicating that co-expressed N, P, and L proteins were functionally active in replication of the agMR derivatives and in transcription of reporter mRNAs.

Here, we updated Figure 6—figure supplement 1 to show detail information of replication and transcription of BYSMV minireplicon. In addition, we insert an A after the start codon to abolish the translation of GFP mRNA, which allow us to observe localization of GFP-P^WT^, GFP-P^S5A^, and GFP-P^S5D^. Use this system, we wanted to show the localization and phase separation of GFP-P^WT^, GFP-P^S5A^, and GFP-P^S5D^ during replication and transcription of BYS-agMR. As expected, GFP-P^S5A^ underwent phase separation and supported high efficient replication of BYSMV. In contrast, GFP-P^S5D^ could not LLPS and supported very less replication of BYSMV. Through RT-qPCR assays, we found that GFP-P^S5A^ and GFP-P^S5D^ support increased replication and transcription, respectively, which is consistent with our previous studies using P^S5A^ and P^S5D^ (Gao et al., 2020). Please see Figure 6—figure supplement 1.

References:

Fang XD, Yan T, Gao Q, Cao Q, Gao DM, Xu WY, Zhang ZJ, Ding ZH, Wang XB. 2019. A cytorhabdovirus phosphoprotein forms mobile inclusions trafficked on the actin/ER network for viral RNA synthesis. Journal of Experimental Botany 70(15): 4049-4062.

Gao Q, et al., 2020. Casein kinase 1 regulates cytorhabdovirus replication and transcription by phosphorylating a phosphoprotein serine-rich motif. The Plant Cell 32(9): 2878-2897.

6) Either soften the claim that viral genomic RNA is recruited to the P-N droplets or include data showing that the full-length BYSMV RNA is indeed recruited to the droplets.

We agree with the reviewers. Because the full-length BYSMV RNA has 12,706 nt and is difficult to be transcribed in vitro, especially using the Cy5-UTP, we cannot show whether the BYSMV genome is recruited into the droplets. We have softened the claim and state that the P-N droplets can recruit the 5′ trailer of BYSMV genome as shown in Figure 3B. Please see line 22, 177 and 190.

7) Remove hollow droplets data of P protein or move the data to supplement.

Removed as suggested. We think that the hollow droplets might be an intermediate form of LLPS. Please see page 7 and page 8.

Reviewer #1 (Recommendations for the authors):In summary, this is an interesting paper. Authors should show that P protein produced during normal infection does indeed undergo phase separation. Authors have necessary tools to demonstrate this. Furthermore, they should by some genetic experiments that NbCK1 is the one that regulates this process.

Thanks for your positive comment and constructive suggestions.

Reviewer #2 (Recommendations for the authors):The authors conclusively demonstrate that P protein phase separates and can be inhibited by CK1 kinase activity. These are the strengths of the study. The conclusions that P phase separation favors replication vs. transcription is not as strongly supported. The minireplicon system data is confusing and the use of 1,6-HD is not sufficient for concluding that phase separation of BYSMV P protein is essential for virus fitness.Below are issues that should be addressed before publication:1. The use of 1,6-hexanediol in Figure 6 is not a suitable approach for multiple reasons.a. 1,6-HD is toxic to cells (see reference in public review).b. 1,6-HD disrupts kinase and phosphatase activity (see ref above).c. The authors do not demonstrate that 1,6-HD inhibited P phase separation in the insect vector or barley (i.e. how do you know the treatment worked?).d. Reduced virus accumulation is likely a result of cell toxicity.

We agree with the reviewer that 1, 6-hexanediol induced cellular toxicity and hereby affected infections of BYSMV and other viruses. Therefore, we removed these results, which does not affect the main conclusion of our results.

2. FRAP data is not properly normalized.a. Figure 1D: The authors state that approximately 80% of the signal recovered following FRAP. However, this is misleading when the photobleaching only reduced the signal to 20%. The actual recovery was closer to 60%. See https://bio-protocol.org/e2525 for an example of FRAP normalization. Typically, the post bleach signal should be set at 0% to give a more discernable recovery. This issue is also present in Figure 2E. 60% of the signal did not recover based on the data present. The FRAP data must be properly normalized. The %recovery in Figure 2E is less than 50% but the images presented in Figure 2D show much higher recovery (this reviewer sees >80% recovery, possibly higher).

Thanks for your constructive suggestions. Using the suggested method, we analyzed the FRAP data again, and updated the Figure 1D and 2E.

3. Minireplicon data must be better described (including the experimental design).a. Most readers will not be familiar with rhabdovirus replication and minireplicon systems. The system needs to be better described.b. It is unclear how S5D can have the highest RFP transcription, but the lowest RFP translation. Is translational repression occurring? The data in panel D does not match the confocal or western blot data concerning RFP expression.c. How was replication/transcription differentiated between 2x35S-driven transcription?

Many thanks for pointing this out! We also noticed the interesting results that have been repeated independently. As shown the illustration of BYSMV-agMR system in Figure 6—figure supplement 1, the relative transcriptional activities of different GFP-P mutants were calculated from the normalized RFP transcript levels relative to the gMR replicate template (RFP mRNA/gMR), because replicating minigenomes are templates for viral transcription.

Since GFP-P^S5D^ supported decreased replication, the ratio of RFP mRNA/gMR increased although the RFP mRNA of GFP-P^S5D^ is not increased. In addition, the foci number of GFP-P^S5D^ is much less than GFP-P^WT^ and GFP-P^S5A^, indicating mRNAs in GFP-P^S5D^ samples may contain aberrant transcripts those cannot be translated the RFP protein. In contrast, mRNAs in GFP-P^S5A^ samples are translated efficiently. These results were in consistent with our previous studies using the free P^WT^, P^S5A^, and P^S5D^.

Reference: Gao Q, et al., 2020. Casein kinase 1 regulates cytorhabdovirus replication and transcription by phosphorylating a phosphoprotein serine-rich motif. The Plant Cell 32(9): 2878-2897.

Reviewer #3 (Recommendations for the authors):Thus, I recommend this paper to publish in your journal, eLife.In the following, I presented a few concerns in regarding to the conclusion of this paper that the authors may want to consider when revising the manuscript. Major concerns:1) Although the authors claimed that "BYSMV P protein forms liquid-like granules through LLPS in vivo", they did not show any phase separation results of BYSMV P protein in barley, the real host of BYSMV. Although they showed some granular structure in Figure S1A (the quality of Figure 1A is bad), they didn't provide evidence to prove these granules are generated by BYSMV. Instead of granules generated by viruses, these granules may be host P bodies or stress granules that react to viral infections. The author can perform immune-electron microscope or other experiments to prove that these granules are indeed formed by P protein.

In addition, we performed immune-electron microscope using anti-P antibodies, showing that the presence of BYSMV P in the viroplasms of BYSMV-infected leaves, rather than mock leaves. Please see Figure 6—figure supplement 1

We used the marker of host P bodies, DCP1-mCherry, to show that DCP1-mCherry and GFP-P were not co-localized in vivo, which has been described in our previous work.

**Author response image 1. sa2fig1:** 

Reference:Zhang Z-J, Gao Q, Fang X-D, Ding Z-H, Gao D-M, Xu W-Y, Cao Q, Qiao J-H, Yang Y-Z, Han CJE. 2020. CCR4, a RNA decay factor, is hijacked by a plant cytorhabdovirus phosphoprotein to facilitate virus replication. eLife 9: e53753.

2) In Figure 6, the author revealed that the phosphorylation of P protein modulates the proliferation of viruses. Although the reviewer believes that the phase separation of P protein facilitates the viral proliferations, the author does not provide direct evidence to prove that. They only showed the correlation of P protein phase separation and viral proliferations. As it is very hard to prove this hypothesis in vitro, the author can lower the tone of their statements.

We agree with the reviewers and softened the claim. Indeed, all negative- stranded RNA viruses are dependent viroplasms for replication. In addition, the P^S5D^ mutant is compromised in phase separation and virus proliferation, indicating that the P phase separation is important for virus replication. Please see Page 11 and 12.

3) By detecting the accumulations of full-length genome minigenome and the mRNA in Figure 6C and 6D, the author claimed that the phase separation of P protein enhances replication but suppress transcription. However, the accumulations of virus transcript may not only be caused by the increased biogenesis, but also may be the results of decreased degradation. Again, the author needs to tone down their conclusion.

We agree with the reviewers and softened the claim. In this work, the GFP-P^WT^, GFP-P^S5A^, and GFP-P^S5D^ were used to show phase separation of P in vivo and virus replication together, which is in consistent with our previous studies using the free P^WT^, P^S5A^, and P^S5D^ (Gao et al., 2020). Please see Page 11 and 12.

Reference: Gao Q, et al., 2020. Casein kinase 1 regulates cytorhabdovirus replication and transcription by phosphorylating a phosphoprotein serine-rich motif. The Plant Cell 32(9): 2878-2897